# Bioactive Compounds and Metabolites from Grapes and Red Wine in Breast Cancer Chemoprevention and Therapy

**DOI:** 10.3390/molecules25153531

**Published:** 2020-08-01

**Authors:** Danielly C. Ferraz da Costa, Luciana Pereira Rangel, Julia Quarti, Ronimara A. Santos, Jerson L. Silva, Eliane Fialho

**Affiliations:** 1Departamento de Nutrição Básica e Experimental, Instituto de Nutrição, Universidade do Estado do Rio de Janeiro, Rio de Janeiro 20550-013, Brazil; daniellyferraz@ymail.com (D.C.F.d.C.); ronimaras@gmail.com (R.A.S.); 2Faculdade de Farmácia, Universidade Federal do Rio de Janeiro, Rio de Janeiro 21941-902, Brazil; lprangel@pharma.ufrj.br; 3Departamento de Nutrição Básica e Experimental, Instituto de Nutrição Josué de Castro, Universidade Federal do Rio de Janeiro, Rio de Janeiro 21941-902, Brazil; julia_quarti@hotmail.com; 4Programa de Biologia Estrutural, Instituto de Bioquímica Médica Leopoldo de Meis, Instituto Nacional de Ciência e Tecnologia de Biologia Estrutural e Bioimagem, Universidade Federal do Rio de Janeiro, Rio de Janeiro 21941-902, Brazil

**Keywords:** bioactive compounds, metabolites, wine, grapes, breast cancer, chemoprevention, chemotherapy

## Abstract

Phytochemicals and their metabolites are not considered essential nutrients in humans, although an increasing number of well-conducted studies are linking their higher intake with a lower incidence of non-communicable diseases, including cancer. This review summarizes the current findings concerning the molecular mechanisms of bioactive compounds from grapes and red wine and their metabolites on breast cancer—the most commonly occurring cancer in women—chemoprevention and treatment. Flavonoid compounds like flavonols, monomeric catechins, proanthocyanidins, anthocyanins, anthocyanidins and non-flavonoid phenolic compounds, such as resveratrol, as well as their metabolites, are discussed with respect to structure and metabolism/bioavailability. In addition, a broad discussion regarding in vitro, in vivo and clinical trials about the chemoprevention and therapy using these molecules is presented.

## 1. Introduction

Breast cancer was ranked as the fifth leading cause of death (627,000 deaths, 6.6%) worldwide in 2018 [1]. During the last decade, new strategies based on the use of dietary chemopreventive agents for breast cancer management have been developed. Several in vitro and in vivo studies have reported the beneficial effects promoted by bioactive compounds from grapes and its derivative products [2]. The positive impact on health has been attributed to its phenolic compounds, such as flavonoids, stilbenes, anthocyanins and other molecules. This review aims to summarize the current findings regarding the role of bioactive compounds from grapes and red wine and their metabolites on breast cancer chemoprevention and treatment by exploring its molecular targets and mechanisms of action [3].

In cancer research, a wide variety of established breast cancer cell lines are used as experimental models. Most of them resemble the different subtypes of breast cancer seen in the clinic. These cell lines offer an infinite supply of a relatively homogeneous cell population that is capable of self-replication in standard cell culture medium and are available through commercial cell banks. The most commonly used breast cancer cell line in the world, MCF-7, was established in 1973 at the Michigan Cancer Foundation. With different molecular characteristics, MDA-MB-231, MDA-MB-453, MDA-MB-468, TD47D, among others, are frequently used in studies, as will be described below [4].

## 2. Anticancer Effects Produced by Grapes and Seed Extracts

Grapes and their derivative products are a rich source of bioactive molecules, including flavonoid compounds (flavonols, monomeric catechins, proanthocyanidins, anthocyanins, anthocyanidins) and non-flavonoid phenolic compounds (resveratrol), as well as their metabolites. Several molecular pathways involved in breast cancer cell signaling and differentiation, cell cycle arrest, apoptosis, and metastasis can be modulated by these compounds, as described below (Figure 1).

A polyphenolic fraction isolated from grape seeds (GSP) containing 50% of procyanidins, 5% of catechin and 6% of epicatechin that has been described to cause irreversible inhibition growth of MDA-MB-468, a metastatic breast cancer cell line, by a mechanism involving activation of MAPK/ERK1/2 and MAPK/p38, the two MAPK pathways associated with cell growth and differentiation. GSP also promoted the induction of CDKI Cip1/p21 and a decrease in CDK4, resulting in G1 arrest [5]. Polyphenols obtained by hydroalcoholic extraction from grape seeds promoted a selective inhibition of cell viability and induction of apoptotic cell death on MCF-7 cells. The authors hypothesize that this effect is mediated by gap-junction-mediated cell-cell communications improvement via re-localization of Cx43 proteins and up-regulation of CX43 gene, since gap junctions have been associated with the apoptotic process [6].

Extracellular matrix remodeling, which is influenced by urokinase-type plasminogen activator (uPA) and matrix metalloproteinases (MMPs), is a critical event in the metastasizing process. Dinicola et al. [7] have studied the effect of grape seed extract (GSE) containing 6.2 mg/g of catechins and 5,6 mg/g of procyanidins on metastatic human breast carcinoma cell line focusing on migration and invasion processes. Low GSE concentrations (25 µg/mL) were reported to strongly inhibit MDA-MB-231 cell migration and invasion by decreasing uPA, MMP-2 and MMP-9 activity, as well as down-regulating β-catenin, fascin and NF-κB expression. On the other hand, high GSE concentration (50 and 100 µg/mL) triggered proliferation arrest and apoptosis.

Another target in the treatment of several cancers is the inhibition of angiogenesis, which is supported by vascular endothelial growth factor (VEGF). A study by Lu et al. [8] showed that GSE (85% procyanidins) reduced VEGF expression in both U251 human glioma cells and MDA-MB-231 human breast cancer cells, supporting the hypothesis that GSE may be a natural anti-angiogenesis source of compounds.

Previous studies described many phytochemicals found in grapes and wine as aromatase inhibitors [9]. Polyphenols were demonstrated to modulate estrogen signaling and to compete for steroid-binding sites. Kijima et al. [10] reported that GSP (74–78% proanthocyanidins and <6% catechin, epicatechin, and their gallic acid esters) suppressed aromatase expression and activity on MCF-7aro (aromatase transfected MCF-7 cells) and SK-BR-3 cells. Aromatase catalyzes critical reactions of estrogen synthesis, converting androgen to estrogen, which is known to stimulate breast cancer cell growth by binding to the estrogen receptor (ER). Another approach includes the combination of therapeutic compounds, like doxorubicin (Dox), and phytochemicals for cancer management. GSE (95% procyanidins) increases the efficacy of Dox in human breast cancer MCF-7, MDA-MB468, and MDA-MB231 cells suggesting a strong possibility of a synergistic effect of GSE and Dox combination, independent of the estrogen receptor status of cells [11].

Grape seed proanthocyanidin extract (GSPE) showed also a promising therapeutic role against adverse effects of the chemotherapeutic agents carboplatin and thalidomide. Administration of these agents in rats led to an enhancement in the TNF-α and IL-6 cytokine levels, which could be partially reversed by administration of GSPE. In addition, GSPE reduced free radicals like thiobarbituric acid-reactive substances and nitric oxide and increased glutathione and antioxidant enzymes in liver and heart [12].

## 3. Flavonoid Compounds

### 3.1. Flavonols

Flavonols are a subgroup of the flavonoids group, structurally resembling flavones, with the presence of an additional hydroxyl in position 3 of the A ring of the flavones general backbone (3-hidroxyflavone) (Figure 2). The most abundant flavonols present in wine are quercetin, a majoritary compound in this beverage, kaempferol, myricetin, isorhamnetin (quercetin 3′-methylether), and rutin, a glucoside derivative of quercetin [13].

In red wine, flavonol (aglycones and glycosides) concentration ranges from around 3 to 50 mg/L [14,15], while quercetin varies from around 1 to 10 mg/L in different wines from different regions of the world [16,17]. Flavonol concentration appears to be related to the degree of sun exposure of the grapes while cultivated and the degradation is both related to UV exposure and temperature [13,18]. Moreover, flavonol content is dramatically altered during the processing of grapes, wine production, and storage [13,19,20].

Flavonol bioavailability may vary according to the substitutions found in the molecule, e.g., sugars (mostly glycosides). A relevant example is rutin, a glycosidic disaccharide conjugate of quercetin [13]. The glycosylated forms are the most frequently found in wine and these compounds appear to be more resistant to degradation than their correspondent aglycones [21].

We discuss below the particular effects of quercetin and kaempferol, the most abundant flavonols in wine, on breast cancer. Their effects encompass both chemoprevention and therapy. Although their aglycone forms share a high structural resemblance, the targets, mechanisms of action and bioavailability of these compounds may vary according to the metabolites produced and substitutions found [22,23].

The anticancer effects of kaempferol and quercetin have been described in several different cancer types, such as bladder, breast, prostate, ovarian, liver, and colon. A special feature is given to breast cancer due to a superior number of published works. The few epidemiological studies available using flavonoids such as quercetin or kaempferol on breast cancer involve the observation of their dietary contribution. However, their findings vary from null to positive effects in cancer prevention [24,25,26]. It is important to mention that, in spite of the larger number of basic research papers published so far, there is a small number of pre-clinical and clinical studies using kaempferol or quercetin as anticancer agents [27,28]. This is due to their low bioavailability, which is around 2% for kaempferol [29,30] and 20% for quercetin, while only 1% was found in the free form in serum [31,32]. This has inspired several works describing drug delivery systems including quercetin and other drugs, such as doxorubicin [33,34,35]. Likewise, the blending of kaempferol and other flavonoids, either in a complex mixture [36] or simply in combination with quercetin, has been described to increase their anticancer properties [37].

In a general way, the anticancer activities of these two flavonols in breast cancer can be organized in three groups: apoptosis induction, growth inhibition (cell cycle arrest), and inhibition of the metastatic behavior: invasion, migration and epithelial-mesenchymal transition (EMT). Several different papers describe the effects of kaempferol and quercetin in the induction of apoptosis in breast cancer cells, such as MCF-7, MDA-MB-453, SK-BR-3, and MDA-MB-231 [38,39,40]. Kaempferol and quercetin decrease the growth of MCF-7 and MDA-MB-231 cells with micromolar concentrations. However, the MCF-7 cell line appeared to be more sensitive to quercetin than MDA-MB-231 [41,42,43,44]. The same was observed for kaempferol in an MCF-7 3-D model, with ERK signaling being responsible for the apoptotic death [45] and cell cycle arrest at the sub-G1 phase [40].

Kaempferol was found to inhibit the RhoA and RacA signaling pathways, leading to cell migration and invasion inhibition in MDA-MB-231 and MDA-MB-453, triple negative breast cancer (TNBC) cells [41]. This compound was also found to promote PARP cleavage for apoptosis induction through the downregulation of Bcl-2 and BAX induction [38]. On the other hand, kaempferol demonstrated a strong antioxidant activity that was able to attenuate ROS-induced hemolysis and to promote an antiproliferative effect on different tumor cell lines, including MCF-7 cells [46].

Quercetin has been widely reported for its activity against breast cancer cell lines such as MCF-7 and MDA-MB-231 [47,48,49]. The mechanisms of action involved apoptosis induction through different pathways, such as caspase activation through the mitochondrial pathway [50,51], inhibition of the Akt signaling pathways [52,53], and cell cycle arrest in the G2/M phase [39,44,50,54]. These effects have been shown in vitro, but also in vivo [22]. Quercetin induces necroptosis, with an increase in BAX expression and Bcl-2 inhibition [55]. In combination with chloramphenicol, isorhamnetin, the main metabolite of quercetin in mammals, was shown to induce mitochondrial fission through CaMKII/Drp1, leading to apoptosis [56]. Also, quercetin-3-*O*-d-galactopyranoside induced apoptosis via ROS through the inhibition of NFκβ signaling pathway and activation of the BAX-caspase 3 axis [57]. Dietary quercetin has demonstrated an inverted “U”-shaped dose-dependent curve on the C3(1)/SV40 Tag breast cancer mouse model [58].

The antimetastatic potential of these compounds involves their ability to inhibit the expression of metalloproteinases such as MMP-3 and MMP-9 [53,59,60]. EMT and angiogenesis in breast cancer cells are also inhibited by quercetin [61,62].

Fatty acid synthase inhibition by flavonoids has been reported and associated with modulation of cell growth and promotion of apoptosis [63]. Quercetin led MDA-MB-231 and MCF-7 cells to EGFR reduction and this signaling promoted fatty acid alterations, including fatty acid isomerization and free radicals production [64].

The structure similarity of estradiol and flavones confers on them a potential to interfere in tumor growth and development through the interaction with ERs [65,66,67,68] and aromatase [69,70]. Preliminary reports show an anti-estrogenic activity for quercetin [71,72], but kaempferol stood out in a panel of phytoestrogens as the one with the most affinity with ERα, which enhances estrogen-dependent cell proliferation [68], displaying estrogenic affinity at 5 µM in a luciferase model in MCF-7 cells. Also, in this cell line, kaempferol revealed a dual effect, according to the concentration used: at the 10 µM range, it was described as an ER competitor with estrogen, since the increase in estrogen concentration was able to impair its functions. On the other hand, at a higher concentration of kaempferol, 100 µM, increments in estrogen concentration were unable to block the kaempferol effect, suggesting different pathways for activation by this compound [73].

In this sense, kaempferol is able to overcome the effects produced by triclosan and bisphenol A, exogenous xenoestrogenic compounds considered endocrine-disrupting chemicals (EDCs) with anti-apoptosis effects [74]. Kaempferol was shown to reverse these effects, increasing BAX levels and reducing Bcl-2 levels, leading VM7Luc4E2 cells to apoptosis [75]. Kaempferol was also able to suppress the EMT and metastatic-related behaviors of MCF-7 cells induced by triclosan [76] and to reverse triclosan-induced phosphorylation of IRS-1, AKT, MEK1/2 and ERK. In a 17β-estradiol (E2) or triclosan tumor growth-induced in vivo xenograft mouse model, co-treatment with kaempferol inhibited tumor growth [74].

Quercetin has been described as a multidrug resistance (MDR) inhibitor in several different works. It is defined as an inhibitor of p-glycoprotein by direct binding to this efflux pump, but also through the downregulation of p-gp expression [77,78,79]. Furthermore, quercetin was shown to potentiate the doxorubicin effect and to reduce its toxicity and side effects, both in vitro and in vivo [33,80]. Similar effects were seen with other drugs such as docetaxel [81], tamoxifen [82], paclitaxel and vincristine [79] using different drug delivery systems. Kaempferol is able to reverse drug resistance promoted by ABCG2 [83,84] and to inhibit quercetin efflux by this transporter [85].

Multidrug transporters are also acknowledged in the metabolization/elimination of quercetin and kaempferol. ABCG2 and ABCC2 participate in kaempferol-3-glucuronide (the major metabolite of kaempferol) elimination in vivo [86]. Finally, the cooperation between ABC transporters and UDP-glucuronosyltransferases appears to regulate kaempferol glucuronidation, thus regulating its accumulation in cells (in comparison to the glucuronide forms) with effects on the pharmacological properties of this compound [87].

### 3.2. Monomeric Catechins and Proanthocyanidins

Catechins, epicatechins, and proanthocyanidins are naturally occurring flavan-3-ols, typically found in tea, cocoa, grape, and wine [88]. Proanthocyanidins are the major phenolic compounds in grape seed and skin [89] and catechins are present in large amounts in green and black teas [90] and in red wine [91]. Proantocyanidins, also known as condensed tannins, are phenolic compounds that take the form of dimers, trimers, and highly polymerized oligomers of flavan-3-ol units [92,93]. Therefore, proanthocyanidins are metabolized to catechins and catechin derivatives [94].

Previous studies indicated that (+)-catechin and (−)-epicatechin (Figure 3) are rapidly absorbed from the upper portion of the small intestine in both human and animal organisms [95]. Catechin bioavailability is inversely proportional to its molecular masses. For example, although the (−)-epigallocatechin-3-gallate (EGCG) content in tea is much higher than other catechins, the peak plasma levels for EGCG (458 Da), (−)-epigallocatechin (306 Da), and (−)-epicatechin (290 Da) are 0.26, 0.48, and 0.19 μM, respectively. Plasma (+)-catechin concentrations increased in response to the ingestion of a single serving of reconstituted red wine. A maximum level of (+)-catechin at 76.7 nmol/L was detected in humans at 1.4 h after intake of both dealcoholized and reconstituted wine [91]. The bioavailability of procyanidins closely resembles that of flavan-3-ol monomers. Different studies, following the ingestion of GSE and GSPE, have shown that during digestion, the oligomers are fragmented into monomeric units of (+)-catechin and (−)-epicatechin and free forms of dimers and trimers have been detected in rat plasma [96,97]. Procyanidin B1 was also detected in human serum 2 h after intake of GSE [98]. Biotransformation of catechins directly undergoes phase II of metabolism, where they can be methylated by catechol-*O*-methyltransferase (COMT), glucuronidated by UDP-glucuronosyltransferase (UGT) or sulfated by sulfotransferase (SULT) [99]. Catechins can also be degraded in the intestinal tract by microorganisms to ring fission metabolites M4, M6, and M6′ [100].

The biological activities of these polyphenols that exert an effect on breast cancer were obtained from a variety of studies, including in vitro and in vivo data. Isolated catechin decreased cell viability and proliferation of MCF-7 human breast cancer cells at 30 and 60 μg/mL [101]. Alshatwi et al. [102] demonstrated that catechin hydrate (150 µg/mL and 300 µg/mL) effectively induced apoptosis in MCF-7 cells through increased expression levels of caspases -3, -8, -9 and p53.

Interestingly, inhibition of cell proliferation by purified (+)-catechin and (−)-epicatechin was more effective in hormone-sensitive breast cancer cells (MCF-7 and T47D), also demonstrating a possible implication of steroid hormone receptors in the action of polyphenols, and in fact, a competition of epicatechin for ER was reported [103].

The anti-carcinogenic activity of wine polyphenols is related to the protection of DNA damages by chemically reactive molecules, such as ROS (reactive oxygen species). The antioxidant effect of purified polyphenols was investigated in three types of breast cancer cells. The treatment with catechin and epicatechin decreased about 80% of ROS production on T47D cells, while no effect was noticed on MDA-MB-231 and MCF-7 cells. The authors attributed the results to different constitutive ROS production between cell lines, hormone receptor spectrum (considering interaction of ROS and these molecules) and limitations of the method [103].

The chemopreventive activity of GSP was demonstrated in an established carcinogen-induced animal model of breast cancer. Adult rats that received 5% GSE (86% proanthocyanidins) showed 44% reduction in the number of DMBA (7,12-dimethylbenz(a)anthracene)-induced mammary tumors [94]. The same effect was observed on a xenograft model using BALB/c nu/nu, athymic, ovariectomized mice carrying MCF-7aro tumors. Mice gavaged with GSE (74–78% proanthocyanidins and <6% catechin, epicatechin and their gallic acid esters) had a 70% reduction in tumor growth, indicating that GSE could suppress aromatase-positive tumors in vivo [10]. On the other hand, diets supplemented with 0.1%, 0.5% and 1.0% of grape seed proanthocyanidins (3.8% of catechin and epicatechin, 96.2% of oligomers and polymers) were not effective in reducing DMBA-induced rat mammary carcinogenesis. The authors attributed the results to the poor absorption of the components and, thus, insufficient amounts being available in the mammary gland to modulate tumorigenesis [104].

In humans, a pilot study with daily doses of grape GSPE failed to decrease plasma estrogens in postmenopausal women [105], despite the same extract exhibit inhibition in cell growth on MCF-7 cells in culture [106]. A double-blind placebo-controlled randomized phase II trial investigated the efficacy of IH636 GSPE in patients with tissue induration, considered a late adverse effect of curative radiotherapy for early breast cancer. In this study, 44 volunteers were given 100 mg of GSPE three times a day orally for six months. The authors considered that the study failed to demonstrate the efficacy of orally-administered GSPE, since there was no significant difference between the groups in terms of external assessments (tissue hardness, breast appearance) or patient self-assessments (breast hardness, pain or tenderness), 12 months post-randomization [107], although the same extract exhibited inhibition in cell growth of MCF-7 cells in culture at 25 mg/L [106].

Additionally, other health effects attributed to GSE and GSPE demonstrated potential to improve antioxidant cell defenses and modulate proinflammatory cytokines, which possibly complement the antitumoral functions of these matrices [108].

### 3.3. Anthocyanins and Anthocyanidins

Anthocyanins are the most abundant flavonoid pigments in young red wines, being responsible for their intense red color [109]. To date, the number of reported types of anthocyanins exceeds 600 [110], but the most common anthocyanidins, aglycone forms of anthocyanins, are cyanidin, pelargonidin, delphinidin, peonidin, petunidin, and malvidin [111] (Figure 4). Anthocyanidins could be immediately metabolized after ingestion of anthocyanins since the β-glucosidase found in intestinal bacteria can easily hydrolyze respective anthocyanins (glycosides) to anthocyanidins (aglycones) [112]. Anthocyanins are known for their apparent poor bioavailability (less to 1–2%). However, presystemic metabolism of these compounds may underestimate their bioavailability if only parent compounds and/or phenolic acid breakdown products are targeted in bioassays. Taking into account the original parent compounds, generated metabolites (from phase I and phase II metabolism and from microbiota-generated) and conjugated products, total bioavailability is much higher than previously credited, after all, anthocyanins are very influential to health [113].

Anthocyanins show a range of antitumor activity in vitro and in vivo, from chemoprevention to chemotherapy. Their potential antitumor effects include antioxidant activities, anti-inflammatory effects, anti-mutagenesis, induction of differentiation and cell cycle arrest, stimulation of apoptosis, autophagy modulation, anti-metastasis, reversion of drug resistance and increasing the sensitivity of cancer cells to chemotherapy [114].

The effects of anthocyanins on breast cancer are some of the most studied, both concerning prevention and treatment of this disease. One of the proposed mechanisms of carcinogenesis is the formation of carcinogen-DNA adducts in target tissues, which is essential to the initiation of chemically induced breast cancer [115]. Singletary et al. [116] evaluated an anthocyanin-rich extract from Concord grapes and the major anthocyanins detected in this extract, delphinidin aglycone and its glucoside, for their capacity to inhibit DNA adduct formation due to the environmental carcinogen benzo[a]pyrene (BP) in a noncancerous, immortalized human breast epithelial cell line (MCF-10F). These authors observed that both grape extract (10 and 20 µg/mL) and isolated compounds (0.6 µM) inhibited BP–DNA adduct formation, through enhancing phase II metabolizing enzymes (GST and NQO1) activities, and suppressed reactive metabolites such as ROS.

Syed et al. [117] also showed that delphinidin, the most common anthocyanidin monomer, could prevent tumor development and malignant progression through inhibition of breast oncogenesis. These authors also assessed experiments with the MCF-10A cell line, a non-tumorigenic mammary epithelial cell line for studying normal breast cell function and transformation, and demonstrated that delphinidin (5 to 40 µM) inhibited HGF-induced early biochemical effects, blocking proliferation and migration of this cell line.

Hepatocyte growth factor (HGF) is produced mainly by mesenchymal cells and acts primarily through its only receptor, c-Met [118]. A variety of cellular responses are activated by c-Met/HGF signaling and mediate critical physiological processes for tumor growth and metastasis in human cancers, including angiogenesis [119], cellular invasion [120,121,122], and morphogenic differentiation [123]. In addition to observing effects on non-tumor breast cells, Syed et al. [117] showed that delphinidin treatment caused growth inhibition of breast cancer cells that express HGF, suggesting that this compound could prevent HGF-mediated activation of signaling pathways implicated in breast cancer.

The chemopreventive effects of delphinidin-3-glucoside were also evaluated on breast carcinogenesis [124]. Yang et al. [124] described that this phytochemical effectively suppressed carcinogenic transformation of MCF-10A cells induced by carcinogen treatment (NNK and BP). After that, these authors investigated the molecular mechanism related to lncRNA HOX transcript antisense RNA (HOTAIR) modulation. Long non-coding RNAs (lncRNA) are usually related to a group of RNAs with more than 200 nucleotides and are not involved in protein generation, despite being involved in different regulatory processes, such as modulation of gene expression [125]. HOTAIR, which is over-expressed in different types of cancers, is a lncRNA that plays a role in carcinogenesis and cancer progression by promoting cancer cell viability, growth, and metastasis [126]. In its turn, HOTAIR is regulated by the interferon regulatory factor-1 (IRF1) protein, which decreases HOTAIR expression. On the other hand, Akt activation decreases IRF1 expression and, consequently, elevates HOTAIR levels [127,128]. Yang et al. [124] observed that delphinidin-3-glucoside treatment (40 µM) inhibited HOTAIR expression in breast carcinogenesis and breast cancer cells. Besides that, these researchers also found the same results in xenografted breast tumors in athymic mice. Mechanistically, in this study delphinidin-3-glucoside down-regulates HOTAIR by inhibiting Akt activation and promoting IRF1.

The first report of tumor cell proliferation inhibitory activity of anthocyanidins from grape skin was published by Zhang et al. [129]. These authors tested the cell proliferation inhibitory activity of five anthocyanidins (cyanidin, delphinidin, pelargonidin, petunidin, and malvidin) and four anthocyanins (cyanidin-3-glucoside, cyanidin-3-galactoside, delphinidin-3-galactoside, and pelargonidin-3-galactoside) against diverse human cancer cell lines. Although anthocyanins did not inhibit proliferation of any cell line tested, even at the highest concentration (200 µg/mL), anthocyanidins inhibited cancer cell proliferation, with malvidin and pelargonidin being the most promising compounds, since they affected many different cancer cells at the same time.

Afaq et al. [130] evaluated the effect of delphinidin (5–40 µM) on epidermal growth factor receptor (EGFR)-positive breast cancer AU-565 cells and non-tumorigenic MCF-10A cells. EGFR is overexpressed in about 20% of invasive breast carcinoma [131] and has been implicated in tumor progression since it promotes a loss of balance between proliferation and apoptosis. The authors showed that this compound is an inhibitor of EGFR and its downstream signaling, the PI3K/AKT and MAPK pathways, both of which play a significant role in the mitogenic and cell survival responses mediated by EGFR. This same study also demonstrated that delphinidin treatment caused more dramatic inhibition of growth of AU-565 cells than MCF-10A cells and had minimal effect on normal mammary epithelial 184A1 cells, which express very low levels of EGFR, suggesting an important contribution of EGFR in delphinidin action. Moreover, delphinidin treatment of AU-565 cells resulted in induction of caspase 3-dependent apoptosis.

Delphinidin has also been shown to induce apoptosis and autophagy in MDA-MB-453 (concentrations of 20, 40 and 80 µM) and BT474 (concentrations of 60, 100 and 140 µM) cell lines [132]. In this study, the autophagy inhibitors, 3-methyladenine (3-MA) or bafilomycin A1 (BA1), enhanced the delphinidin-induced apoptosis in both breast cancer cell lines, suggesting that autophagy might exert a protective effect in this experimental model. In addition, these authors showed that delphinidin induced autophagy via the mTOR and AMPK signaling pathways.

The effect of cyanidin-3-glucoside, the main anthocyanin studied in breast cancer cells, was evaluated on breast cancer-induced angiogenesis [133]. This anthocyanin attenuated breast cancer-induced angiogenesis via inhibiting the expression and secretion of VEGF, the most important angiogenic cytokine, in a dose-dependent manner (concentrations up to 20 µM). The mechanism proposed by these authors involves the downregulation of STAT3, at both mRNA and protein level, via inducing miR-124, resulting in VEGF inhibition. Thus, the inhibitory effect of cyanidin-3-glucoside on the endogenous STAT3 may occur in a non-canonical way, via miRNAs, which could downregulate target gene expression with mRNA degradation.

Recently, Liang et al. [134] reported that cyanidin-3-glucoside (20 µM) decreased the migratory and invasive nature of triple-negative breast cancer cell lines through reversion of the EMT, which is highly associated with cancer metastasis. Cyanidin-3-glucoside increased epithelial markers (E-cadherin and zonula occludens-1), decreased mesenchymal markers (vimentin and N-cadherin) and EMT-associated transcription factors (Snail1, Snail2). Mechanistically, this phytochemical also attenuated the pivotal factor for EMT, NF-κB, and induced the inhibitor Sirt1 in triple-negative breast cancer cell lines.

There is currently available evidence that endogenous estrogens play a critical role in the development of breast cancer [135]. Despite the fact that triple-negative breast cancer is characterized by a lack of ERα expression [136], data suggest that estrogen still plays a critical role in the etiology of this type of cancer since a 36-kDa variant of ERα, known as ER alpha 36 (ERα36), is highly expressed in triple-negative breast cancer [137] and is involved in rapid estrogen signaling [138]. Studies also emphasize the causal link between ERα36 and EGFR, since this receptor is one of the most critical downstream targets of activated ERα36 signaling [138].

Wang et al. [139] found that cyanidin-3-glucoside (150 µM) preferentially promotes cell death, by the extrinsic apoptosis pathway of triple-negative breast cancer cells (MDA-MB-231) that co-express ERα36 and EGFR. Cyanidin-3-glucoside directly binds to the ERα36 receptor, which in turn inhibits its downstream signaling, the EGFR/AKT pathway leading to EGFR degradation through the proteasome system. A xenograft mouse model also confirmed these properties of cyanidin-3-glucoside.

Fernandes et al. [140] evaluated the effect of cyanidin-3-glucoside, delphinidin-3-glucoside, and vinylpyranoanthocyanin-catechins (portisins) on MCF-7 cells. Overall, the studied compounds inhibited, in a dose-dependent manner (12.5–100 µM), the growth of MCF-7 cells, however, delphinidin-3-glucoside and its respective portisin presented the highest cytotoxic effect. This same study also highlighted a structural requirement for a more potent cytotoxicity effect on MCF-7 cells, characterized by an ortho- trihydroxylated substituent attached to the phenolic ring. Nevertheless, this study was unable to elucidate whether anthocyanins antiproliferative effect could be dependent or independent of ERs or other molecular pathways involved.

Although the monoclonal antibody trastuzumab improves survival of patients with HER2-positive breast cancers [141], the majority of patients who initially respond to this therapy demonstrate disease progression within 12–24 months [142]. Thereby, identifying alternative strategies to overcome trastuzumab resistance targeting HER2 may improve treatment response in breast cancer. Li et al. [143] investigated the antitumor properties of anthocyanins, peonidin-3-glucoside, and cyanidin-3-glucoside, on parental HER2-positive cells and their trastuzumab-resistant cell lines. Treatment with cyanidin-3-glucoside and peonidin-3-glucoside significantly inhibited cell growth in the parental and trastuzumab-resistant cells in a dose-dependent manner. The authors also observed that the mechanisms of action of cyanidin-3-glucoside (5 μg/mL) and peonidin-3-glucoside (5 μg/mL) involve the inhibition of HER2 phosphorylation, the induction of apoptosis in both sensitive and trastuzumab-resistant cell lines, and in a higher anthocyanins concentration (1 mg/mL), an inhibition of trastuzumab-resistant cells migration and invasion was also observed. Besides that, treatment with both anthocyanins (6 mg/Kg/twice weekly, intraperitoneal) reduced trastuzumab-resistant cell-mediated tumor growth in vivo.

As described above, cyanidin-3-glucoside can act alone against breast cancer. However, it has also been shown to be effective in combination with trastuzumab in three representative HER2-positive breast cancer cell lines [144]. These researchers demonstrated that HER2 inactivation seems to represent a central role in the synergistic effect between cyanidin-3-glucoside and trastuzumab in all HER2-positive breast cancer cells tested. Moreover, cyanidin-3-glucoside (5 μg/mL) alone and in combination with trastuzumab (5 μg/mL) induced cell apoptosis in HER2-positive cell lines. These authors also evaluated, in an in vivo xenograft model in mice, the effect of 6 mg/mL cyanidin-3-glucoside in association with 6 mg/mL trastuzumab intraperitoneally twice a week for 25 days. These results demonstrated that anthocyanins were able to significantly enhance trastuzumab-induced tumor growth inhibition.

## 4. Non-Flavonoid Phenolic Compounds

### Resveratrol

Natural stilbenes are an important group of non-flavonoid polyphenols characterized by the presence of a 1,2-diphenylethylene nucleus in their structure [145]. Among them, resveratrol (3,4′,5-trihydroxy-*trans*-stilbene) is a phenolic compound derived from grapes, berries, peanuts, and other plant sources. The molecule consists of two aromatic rings that are connected through a methylenic bridge and exists as *cis*- and *trans*-resveratrol isomers (Figure 5), and their glucosides, *cis*- and *trans*-piceid [146]. Resveratrol was originally identified as a phytoalexin by Langcake and Pryce [147] and is produced by a wide range of plant species under stressful environmental conditions, such as pathogen infection and ultraviolet radiation. Grapes and their derivative products, particularly red wine, are the most important natural sources of resveratrol. The resveratrol composition of wines depends on the grape varieties used, as well as the growing conditions and the wine-making methods, which may vary. In fresh grape skin, the concentration of this compound is in the range of 50–100 µg per gram, and red wine contains about 1.9 ± 1.7 mg/L of *trans*-resveratrol [148,149,150,151,152].

In humans, resveratrol is extensively metabolized and rapidly eliminated. When consumed orally, the molecule is absorbed via passive diffusion or by membrane transporters in the intestine, and then conjugated into glucuronides and sulfates. Although oral absorption is around 75%, only a small fraction of resveratrol ingested from dietary sources reaches the bloodstream and body tissues. It was previously described that metabolism in the liver and intestine results in oral bioavailability of about 1–2% of *trans*-resveratrol [153,154,155]. Rapid conjugation and low bioavailability are some of the major limitations and challenges of the in vivo use of this compound. Different methodological approaches, such as encapsulation in liposomes, emulsions, micelles, insertion into polymeric nanoparticles, solid dispersions, and nanocrystals, have been developed to improve the low aqueous solubility and the poor bioavailability of resveratrol [156]. Furthermore, the use of naturally occurring or synthetic resveratrol derivatives, with a better pharmacokinetic profile, low toxicity, less side effects, and improved biological activities, are promising strategies for clinical applications of stilbene compounds [157].

Resveratrol exhibits multiple bioactivities, including anti-oxidative, anti-inflammatory, cardioprotective, neuroprotective, anti-aging and anticancer properties. Accumulated experimental and clinical evidence clearly shows the chemopreventive and chemotherapeutic potential of resveratrol, as reviewed in our recent publication [158]. Scientific interest in this molecule has grown considerably during the last 23 years, since Jang and colleagues first demonstrated the ability of resveratrol to inhibit in vivo carcinogenesis in a mouse skin cancer model [159,160]. Resveratrol is reported to act as a multi-target suppressor of all three carcinogenesis stages (initiation, promotion, and progression), by regulating signal transduction pathways that control cell division and growth, apoptosis, inflammation, angiogenesis, and metastasis. Furthermore, resveratrol increases the efficacy of traditional chemotherapy and radiotherapy by reducing drug resistance and sensitizing tumor cells to a chemotherapeutic agent [160,161,162,163].

A plethora of studies, including in vitro and in vivo investigations, have suggested that resveratrol triggers chemopreventive and therapeutic responses against several tumor types, such as skin, breast, prostate, lung, colon, and liver cancer [163,164]. As indicated by a recent search on PubMed (accessed in April 2020), most of these studies (570 of 3524 hits) have been reported in breast cancer models. In 2005, it was shown for the first time that resveratrol from grape consumption is inversely related to breast cancer risk, as reported in a case-control study conducted between 1993 and 2003 in the Swiss Canton of Vaud on 369 cases and 602 controls [165]. Among its wide range of biological properties, resveratrol has attracted considerable attention in breast carcinogenesis because of its role as a phytoestrogen. This compound can compete with natural estrogens for binding to ERs, thus modulating its biological responses [146,155,166,167].

Hormone-dependent tumors may be prevented by regular exposition to selective estrogen receptor modulators (SERMs). These compounds exhibit different levels of estrogen agonism or antagonism, depending on the cell type and gene expression targeted by ERs [168]. Gehm and colleagues were the first to investigate whether resveratrol would have estrogenic activity due to its structural similarity to the synthetic estrogen diethylstilbestrol (DES; 4,4’-dihydroxy-*trans*-α, β-diethylstilbene). Based on its ability to compete with E2 for binding to and modulating the activity of ERα, resveratrol was characterized as a phytoestrogen [169]. It binds to ER at a low micromolar range (3–10 µM) and with lower affinity than estradiol. Despite this, resveratrol may act as a superagonist in activating hormone receptor-mediated gene transcription in MCF-7 cells [169,170]. In contrast, the antiestrogenic activity of resveratrol in breast cancer was also reported, being related to pathways that inhibit estrogen-induced cellular outcomes, such as proliferation, tumoral transformation, and progression [146]. Lu and Serrero reported ER antagonism of resveratrol (5 µM) in the presence of E2 and partial agonism in its absence [171]. It was also demonstrated that resveratrol exerted a mixed agonist/antagonist action in cells transiently transfected with ER, and mediated higher transcriptional activity when bound to ERβ than to ERα. Moreover, resveratrol showed antagonist activity with ERα, but not with ERβ [172]. Based on these reports, resveratrol may be categorized as a natural SERM, since it behaves as both agonist and antagonist of ERs. These opposite responses vary according to cell type, resveratrol concentration, hormone competition and ERs expression [155,173]. Resveratrol also modulates the expression of the progesterone receptor (PR). It was previously reported that resveratrol produces greater transcriptional activation of PR than estradiol. In MCF-7 cells, resveratrol was as effective as a maximal dose of estradiol in activating PR gene expression [169].

In tumors, expression of aromatase is upregulated compared to that of surrounding noncancerous tissue. The suppression of in situ estrogen formation by using aromatase inhibitors is a promising route for chemoprevention against breast cancer. In SK-BR-3 cells, resveratrol significantly reduced aromatase mRNA and protein expression in a dose-dependent manner [174]. Resveratrol also inhibits the expression and enzyme activity of aromatase, thus reducing localized estrogen production in breast cancer cells [175]. When tested in a co-culture system of T47D breast cancer cells with human breast adipose fibroblasts (BAFs), resveratrol (20 µM) promoted an aromatase inhibitory effect as potent as 20 nM of letrozole, which is a clinically used anti-aromatase drug in breast cancer treatment [176].

As reviewed by different authors, several experimental approaches have been used to describe the molecular mechanisms of resveratrol in breast carcinogenesis [155,158,162,177]. In addition to the phytoestrogenic action, resveratrol modulates xenobiotic metabolism by altering ABCG2 and CYP1A1 activities [178]; decreases the production of prostaglandins by inhibiting COX-2 expression and activity at multiple levels [177]; suppresses the growth of different breast cancer cell lines and induces a number of biological pathways, thus leading to cell growth arrest and apoptosis [155,165,177,179,180]; modulates the p53 tumor suppressor protein by inducing post-translational modifications [158,180]; prevents mutant p53 aggregation in breast cancer cells and in breast tumor xenografts [181]; regulates extracellular growth factors and receptor tyrosine kinases [162]; induces epigenetic mechanisms by modulation of histone acetylation/methylation [182]; inhibits angiogenesis, EMT, and metastasis [155]; acts as an MDR reversion molecule [183] and sensitizes breast cancer cells toward chemotherapy [161]. In animal studies, resveratrol inhibits chemically-induced breast carcinogenesis; it reduces tumor growth, decreases angiogenesis and increases the apoptotic index in xenograft breast cancer models; delays the tumor development, reduces the mean number and size of tumors and diminished the number of lung metastases in spontaneous breast tumor models [155]. In recent years, accumulating evidence also suggests that resveratrol may be effective in breast cancer management when given in combination with other naturally occurring and chemotherapeutic agents, thus suggesting that resveratrol can enhance the efficacy of other compounds [184].

Although the antitumor activity of resveratrol in in vitro and animal breast cancer models is well established, the clinical evidence regarding its therapeutic effect against breast cancer is still limited. Considering that preclinical and clinical studies suggested that resveratrol may modulate several hormone-related factors involved in breast cancer risk, a pilot phase I clinical study was conducted in a group of forty postmenopausal women with high body mass index, to determine the clinical effect of resveratrol on systemic sex steroid hormones. The resveratrol intervention (1 g daily, for 12 weeks) did not result in significant changes in serum concentrations of estradiol, estrone, or testosterone, but had favorable effects on estrogen metabolism and steroid hormone-binding globulin (SHBG) [185]. Further clinical trials are required to ascertain and validate the efficacy of resveratrol on breast cancer.

## 5. Grape and Wine Metabolites and Breast Cancer (In Vitro and In Vivo Studies)

The health-promoting effect of wine can be focused on consumption, bioavailability, metabolism and microbiota influence on bioactive compounds. There is now strong evidence that the molecules responsible for those effects are probably not the ingested ones but rather their metabolites that occur after the action of microbiota and absorption process. The identification and quantification of these metabolites has not been an easy task, but improvement of analytical methods and sensitivity has allowed some advances in metabolomics area [186].

The WinMet database contains 2030 putative compounds present in oenological matrices covering 10 different families, such as phenols, organic acids, biogenic amines, sugars, polyols, fatty acids, higher alcohols, aldehydes, lignans, and ketones [187]. These molecules can be divided into primary metabolites (e.g., sugars, amino acids, and short chain organic acids) and secondary metabolites (flavonoids and phenol compounds) and are well documented in the literature about wines [188].

Wine is a complex mixture of many different molecules and several factors interfere in its composition, such as grape type, fermentation process, aging, among others. For example, catechin and epicatechin decrease during aging in all wines, while gallic acid increases in almost all red wines [189]. Thus, the purpose of this section will be to discuss in vitro and in vivo studies related to the metabolites of the flavonoid and non-flavonoid compounds present in red wine described previously in this review, and the relationship between these molecules and breast cancer.

The majority of phenolic compounds from grapes and wine are metabolized in the gastrointestinal tract, where they are broken down by gut microbiota and typically involve deglycosylation, followed by breakdown of ring structures to produce phenolic acids and aldehydes. These metabolites can be detected in bloodstream, urine, and fecal samples by using sophisticated instrumentation methods for quantitation and identification at low concentrations [190].

An intervention study with red wine offered to eight healthy adults for 20 days revealed significant changes in eight metabolites: 3,5-dihydroxybenzoic acid, 3-*O*-methylgallic acid, p-coumaric acid, phenylpropionic acid, protocatechuic acid, vanillic acid, syringic acid and 4-hydroxy-5-(phenyl)valeric acid without any influence of ethanol on the microbial action [191]. The same research group characterized the metabolome of human feces after moderate consumption of red wine by healthy subjects during four weeks and showed 37 metabolites related to wine intake, from which 20 could be tentatively or completely identified, including the following: wine compounds, microbial-derived metabolites of wine polyphenols and endogenous metabolites and/or others derived from different nutrient pathways. After wine consumption, fecal metabolome is usually enriched in flavan-3-ols metabolites [192].

To determine which compounds in grapes and wine are the most bioactive, their effects in disease models must be known, including absorption and metabolism. Rats that consume a red wine extract have elevated levels of the microbial phenolic acid metabolites 3-hydroxyphenylpropionic, 3-hydroxybenzoic, 3-hydroxyhippuric, hippuric, p-coumaric, vanillic, 4-hydroxybenzoic, and 3-hydroxyphenylacetic acids in urine. These urine metabolites account for roughly 10% of the administered red wine polyphenols [193]. Most grape and wine flavonoids and others are rapidly metabolized in the human body, making it difficult to determine whether these compounds are effective against disease.

Based on these metabolites, the combination of hippuric acid (HA) nanocomposite (intercalation of hippuric acid into a zinc-layered hydroxide) with doxorubicin and oxaliplatin induced cytotoxicity in MDA-MB-231 and MCF-7 cell lines [194]. 4-Hydroxybenzoic acid (4-HBA) and a histone deacetylase 6 (HDAC6) inhibitor could successfully reverse adryamicin (ADM) resistance in human breast cancer cells. 4-HBA significantly promoted the anticancer effect of ADM on apoptosis induction, as evidenced by the increased expressions of caspase-3 and PARP cleavage, which were associated with the promotion of p53 and homeodomain interacting protein kinase-2 (HIPK2) expressions in ADM-resistant breast cancer cells. Therefore, 4-HBA could be applied as an effective HDAC6 inhibitor to reverse human breast cancer resistance. Herein, the 4-HBA and ADM combination might represent a useful therapeutic strategy to prevent human breast cancer [195].

Apoptotic effects of protocatechuic acid (PCA), another metabolite of wine, were examined on MCF-7 cells. Results showed that PCA concentration-dependently decreased cell viability, increased lactate dehydrogenase leakage, enhanced DNA fragmentation, reduced mitochondrial membrane potential and lowered Na^+^-K^+^-ATPase activity. PCA also concentration-dependently elevated caspases-3 and -8 activities and significantly inhibited cell adhesion. These findings suggest that PCA is a potent anticancer agent to cause apoptosis or retard invasion and metastasis in breast cancer and other cells [196].

The metabolites gallic acid, 4-*O*-methylgallic acid and 3-*O*-methylgallic acid are detected in the plasma of humans who consume 300 mL of red wine [197]. In fact, the metabolites gallic acid and 4-*O*-methylgallic acid are well correlated with wine consumption and may be used as urinary biomarkers for wine intake in health-related studies [198]. Phenolic acid metabolites are mainly formed from gut microbiota metabolism and could be responsible for much of the disease reduction associated with consuming wine and grape phenolics.

Gallic acid (GA) possesses potential for antitumoral activity on different types of malignancies. GA treatment significantly decreased the cell viability of MDA-MB-231 and HS578T cells in a dose-dependent manner. In addition, GA exerted relatively lower cytotoxicity on non-cancer breast fibroblast MCF-10F. The changes in cell cycle distribution in response to GA treatment led to an increase of G0/G1 and sub-G1 phase ratio in MDA-MB-231 cells. GA also downregulated cyclin D1/CDK4 and cyclin E/CDK2, upregulated p21Cip1 and p27Kip1 and induced activation of caspase-9 and caspase-3. In addition, it modulated p38 mitogen-activated protein kinase that was involved in the GA-mediated cell-cycle arrest and apoptosis. GA inhibited the cell viability of TNBC cells, which may be related to the G1 phase arrest and cellular apoptosis via p38 mitogen-activated protein kinase/p21/p27 axis. Thus, GA could be beneficial for TNBC treatment [199]. GA also promoted inhibition of proliferation and induction of apoptosis in MCF-7 cells. The results revealed that GA induced apoptosis by triggering the extrinsic or Fas/FasL pathway as well as the intrinsic or mitochondrial pathway. Furthermore, the apoptotic signaling induced by GA was amplified by a cross-link between the two pathways. Taken together, these findings may be useful for understanding the mechanism of action of GA on breast cancer cells and provide new insights into the possible application of such a compound and its derivatives in breast cancer therapy [200].

### 5.1. Resveratrol Metabolites

Resveratrol is a minor component of red wines and, following its ingestion, it is converted to glucuronide and sulfate metabolites, which are present in the circulatory system in nanomolar concentrations [201]. Nevertheless, the by far most commonly studied form of resveratrol is the aglycone, often at concentrations largely exceeding those attainable in vivo. By contrast, very little is known about the biological activity of the resveratrol metabolites formed upon intestinal absorption, which represent the major circulating forms of resveratrol; in particular, the glucuronic acid and the sulfate conjugates of trans-resveratrol, which are produced at the enterocyte and hepatocyte level [202]. Besides dihydroresveratrol, Bode et al. [203] found, in vivo and in vitro, bacterial trans-resveratrol metabolites: 3′,4-dihydroxy-trans-stilbene and 3′,4′-dihydroxybibenzyl (lunularin). In estrogen-sensitive cancer cells, like MCF-7, 3′,4-dihydroxy-trans-stilbene showed agonist properties [204].

Resveratrol-3-*O*-sulfate (R3S), but no other resveratrol derivative, exerted a pronounced antiestrogenic activity on both receptors (α and β), with a marked preference for ER. R3S, the main resveratrol metabolite accumulating in human plasma after ingestion of dietary amounts of resveratrol, is an effective ER-preferential anti-estrogen in both yeast and mammalian cells [205]. A significant increase in MCF-7 cancer cells growth rates was shown in the presence of picomolar concentrations of dihydroresveratrol (DH-RSV) because this polyphenol has a profound proliferative effect on hormone-sensitive tumor cell lines such as MCF-7.

The proliferative effect of DH-RSV was not observed in cell lines that do not express hormone receptors (MDA-MB-231, BT-474 and K-562) [206]. Human MCF-7 (wild-type p53), MDA-MB-231 (mutant p53) and nontumorigenic MCF-10A cells are treated with resveratrol and physiological-derived metabolites (RSV-3-*O*-glucuronide, RSV-3-*O*-sulfate, RSV-4′-*O*-sulfate, DH-RSV and DH-RSV-3-*O*-glucuronide). Cellular senescence is measured by SA-β-gal activity and senescence-associated markers (p53, p21Cip1/Waf1, p16INK4a and phosphorylation status of retinoblastoma (pRb/tRb). While no effect is observed in MDA-MB-231 and normal cells, resveratrol metabolites induce cellular senescence in MCF-7 cells by reducing their clonogenic capacity and arresting cell cycle at the G2M/S phase, but do not induce apoptosis. Senescence is induced through the p53/p21Cip1/Waf1 and p16INK4a/Rb pathways, depending on the resveratrol metabolite, and requires ABC transporters, but not ERs. Recent evidence demonstrates that resveratrol metabolites, but not free resveratrol, reach malignant tumors (MT) in breast cancer (BC) patients. Since these metabolites, as detected in MT, do not exert short-term antiproliferative or estrogenic/antiestrogenic activities, long-term tumor senescent chemoprevention has been hypothesized. These data suggest that resveratrol metabolites, as found in MT from BC patients, are not deconjugated to release free resveratrol, but enter the cells and may exert long-term tumor-senescent chemoprevention [207].

### 5.2. Catechins Metabolites

Catechin appears to be metabolized only if absorbed from the small intestinal lumen. Both 3′-*O*-methylcatechin-glucuronide and catechin-glucuronide are produced in intestinal cells and methylation and sulfation of catechin metabolites occur in the liver [208]. Catechin glucuronide and 3′-*O*-methylcatechin glucuronide are mainly found in plasma of rats after ingestion of catechins [208,209]. Large amounts of the 3′-*O*-methyl metabolite are also found to be glucuronidated and sulfated on the same compound, presumably produced in the liver, and are only detected in the bile [208]. In humans, between 3.0 and 10.3% of ingested catechins from red wine are accounted for in urine, mostly as catechin and its 39-*O*-methyl-glucuronide and sulfate metabolites [210].

Aside from metabolism that occurs in intestinal cells and liver, catechins can also be metabolized by gut microbiota to produce phenolic acid metabolites. In rats, these metabolites can be found in urine, being 3-hydroxyphenylpropionic acid, 3-hydroxybenzoic acid and 3-hydroxyhippuric acid present in the highest concentrations [193]. When catechin is incubated with human gut microbiota, it is metabolized to 4-hydroxybenzoic acid, 2,4,6-trihydroxybenzaldehyde, phloroglucinol and 4-methoxysalicylic acid [211], again emphasizing the effects of individual microbiota profiles on gut metabolism. We have not found much research showing the association of catechin metabolites with breast cancer, only with the use of phloroglucinol, as can be seen above.

Metastasis is a challenging clinical problem and the primary cause of death in breast cancer patients. Treatment with phloroglucinol (PG) effectively inhibited mesenchymal phenotypes of basal type breast cancer cells through downregulation of SLUG without causing a cytotoxic effect. Importantly, PG decreased SLUG through inhibition of PI3K⁄AKT and RAS⁄RAF-1⁄ERK signaling. Treatment with PG sensitized breast cancer cells to anticancer drugs such as cisplatin, etoposide, and taxol, as well as to ionizing radiation. Taken together, these data indicate PG to be a good candidate to target breast cancer stem-like cells (BCSCs) and to prevent disease relapse [212,213].

### 5.3. Anthocyanins Metabolites

In humans, nanomolar plasma concentrations of anthocyanins are found after they are consumed. Anthocyanins such as cyanidin-3-glucoside and pelargonidin-3-glucoside could be absorbed in their intact form into the gastrointestinal wall, undergo extensive first-pass metabolism, and enter the systemic circulation as metabolites. Phenolic acid metabolites were found in the bloodstream in much higher concentrations than their parent compounds. These metabolites could be responsible for the health benefits associated with anthocyanins [214].

After rats ate cyanidin-3-glucoside, the aglycone was only found in the small intestine, cyanidin-3-glucoside was found in the plasma, and methylated cyanidin-3-glucoside was found in the liver and kidney [215]. Cyanidin-3-glucoside attenuates the angiogenesis of breast cancer via inhibiting STAT3/VEGF pathway [133].

In humans and Caco-2 cells, cyanidin-3-*O*-glucoside’s major metabolites are protocatechuic acid (PCA) and phloroglucinaldehyde which are also subjected to entero-hepatic recycling, although caffeic acid and peonidin-3-glucoside seem to be strictly produced in the large bowel and renal tissues [216].

Previous studies evaluated the bioavailability of anthocyanins using red wine and dealcoholized red wine [217,218]. One of the first studies is the work of Bub and co-workers who only detected the main native anthocyanin in plasma and urine with no effect of ethanol on the amount quantified [217]. Ethanol enhances cyanidin-3-*O*-glucoside’s metabolism potentiating its conversion into methylated and glucuronidated derivatives, showing an increase in the two main anthocyanin conjugates, methyl-cyanidin-glucuronide and 3′-methyl-cyanidin-3-*O*-glucoside, of 59% and 57%, respectively. But in this case, the food matrix used was blackberry puree with or without ethanol, and not wine or grapes [219].

The accumulation of multiple phenolic metabolites might ultimately be responsible for reported anthocyanin bioactivity, with the gut microbiota apparently playing an important role in the biotransformation process. Nevertheless, phase II conjugates of cyanidin-3-*O*-glucoside and cyanidin (cyanidin-glucuronide, methyl cyanidin and methyl-cyanidin-glucuronide) were also detected in plasma and urine [186]. The most important metabolites corresponded to products of anthocyanin degradation (i.e., benzoic, phenylacetic and phenylpropenoic acids, phenolic aldehydes and hippuric acid) and their phase II conjugates, which were found at 60- and 45-fold higher concentrations than their parent compounds in urine and plasma, respectively [220].

Delphinidin-3-glucoside, cyanidin-3-glucoside and petunidin-3-glucoside methylated metabolites were obtained by enzymatic hemi-synthesis and decreased or did not alter the antiproliferative effect of the original anthocyanin in MCF-7 cells [221]. The methylation reaction alters the number of hydroxyl and methoxyl groups in ring B, so these metabolites are likely to have different antioxidant activities in comparison with the original anthocyanins. Generally, the health effects of anthocyanins are associated with an increase in the endogenous antioxidant defenses. In a paper by Fernandes et al. [221] the synthetized methylated metabolites still displayed some antiproliferative activities for the three cell lines although not as intense as parental anthocyanin. The biological studies conducted with the metabolites in comparison with the native compounds allow understanding of the real contribution of methylation towards the antioxidant and antiproliferative effects of anthocyanins. However, this subject is new and needs more publications for a good discussion, especially from methylated anthocyanin-derived metabolites.

### 5.4. Quercetin Metabolites

Quercetin or its metabolites may have cytotoxic activities [222]. Studies on the metabolism of quercetin suggest degradation by intestinal microbiota and relatively low absorption [223], limiting its use as a biomarker. Metabolism of quercetin includes 3,4-dihydroxyphenylacetic acid (DHPAA) as homoprotocatechuic acid, m-hydroxyphenylacetic acid (mHPAA), and 3-methoxy-4-hydroxyphenylacetic acid as homovanillic acid (HVA) [224]. These three metabolites are excreted in the urine of rats, rabbits, and humans [224,225].

Recently, Yamazaki et al. [226,227] investigated the effects of quercetin and its main circulating metabolite quercetin-3-*O*-glucuronide on MCF-10A and MDA-MB-231 cells and suggested that these flavonoids may suppress invasion of these cells by controlling β_2_-adrenergic signaling, and may be a dietary chemopreventive factor for stress-related breast cancer.

### 5.5. Metabolites and Breast Cancer Patients

In relation to breast cancer patients, previous reports have shown that glycolysis, lipogenesis and the production of volatile organic metabolites were increased in the serum of these patients compared to healthy women [228]. The serum levels of choline, tyrosine, valine, lactate, isoleucine are up-regulated, and glutamate levels are downregulated in patients with early-stage breast cancer [229]. These studies reveal that metabolic alterations are important indications for breast cancer. There is evidence that metabolic changes are correlated with metastasis and metabolism of tumors [230,231,232]. Metabolism changes are often associated with resistance to chemotherapy and therapeutic sensitivity in clinical chemotherapy. Breast cancer cells not only show significant differences in metabolism compared with healthy breast cells, but also show differences in drug resistance [233,234]. Cancer and metabolism are deeply interconnected, studies indicate that cancer evolution is associated with abnormal glucose metabolism that is related to high proliferation, metastasis and clinical characteristics and is allied to the action of a particular drug. In this context, chemoresistance enables cancer cells to survive drug action and proliferate uncontrollably, which may lead to strong metastatic potential and cancer progression [230,231,232,233,234].

A recent clinical trial has reported resveratrol accumulation, mainly as sulfates and glucuronides, in normal and malignant human breast tissues. Although phase-II conjugation might hamper a direct anticancer activity, long-term tumor-senescent chemoprevention cannot be discarded [235]. Metabolites of wine bioactive compounds have been positively related to in vitro and in vivo breast anticancer properties and this evidence was associated with the ingestion of several flavonoids present in large amounts in red wine. However, the concentration required to trigger a biological event is dependent not only on the amount ingested, but also on critical variables that include bioaccessibility, bioavailability, stability under in vivo conditions, and so on. Many studies are still required to clarify the role of many of these metabolites with regard to the health-promoting properties of wine.

Table 1 summarizes the data collected from the literature about the metabolite dosage used or found in the different in vitro and in vivo models mentioned in this review.

## 6. Conclusions

As can be seen in this compilation, grapes and wines are rich and complex sources of bioactive molecules with multiple targets and effects. The natural polyphenols from these dietary products belong to different classes of compounds, both flavonoid and non-flavonoid, and have been studied in different models of breast cancer, both in vivo and in vitro. The major anticancer activities promoted by these compounds are summarized in Figure 1 and include modulation of estrogen cell signaling, cancer cell differentiation, cell growth inhibition, apoptosis induction and suppression of the metastatic behavior.

Based on dietary source, bioactive compounds or their metabolites used in different in vitro and in vivo studies for breast cancer, we conclude that there is a great variation of doses utilized or found. When the studies utilize wine or grape as a bioactive compound source, it is possible to observe a great variation on metabolite quality and quantity. On the other hand, when the isolated metabolite or its precursor were used, mainly in cancer cell lines, variations from 1 nM until 100 µM were used, and some authors justify the use of these concentrations to approximate the physiological concentrations. It is also important to point out that the effects produced by the glycosidic forms and the aglycones might lead to different routes of absorption and/or metabolization, leading to important variations in bioavailability and global effects produced.

The bioavailability of these compounds is another important issue that must be circumvented to improve local biological effects. In this way, grape and wine have long been used as sources of lead compounds in the search for breast cancer chemotherapy candidates and should be further explored in clinical studies, along with the biotechnological improvements necessary for their application.

## Figures and Tables

**Figure 1 molecules-25-03531-f001:**
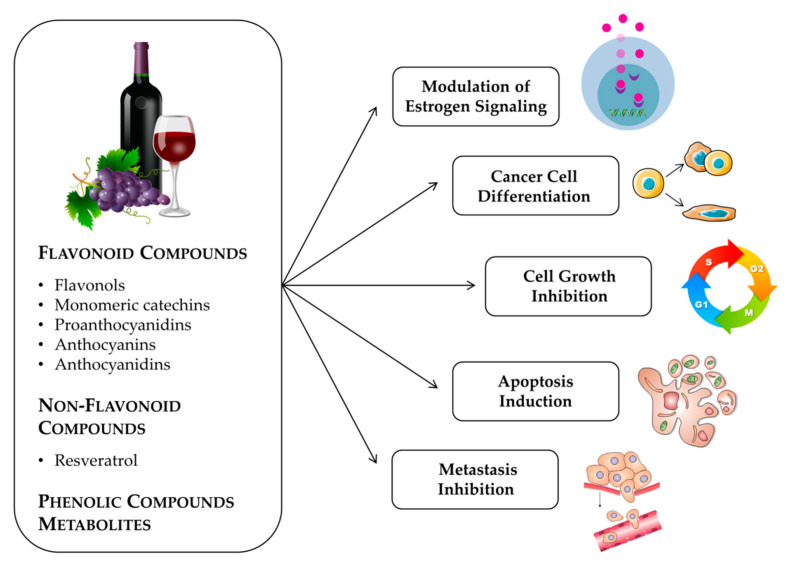
Anticancer activities promoted by phenolic compounds from grapes and red wine and their metabolites.

**Figure 2 molecules-25-03531-f002:**
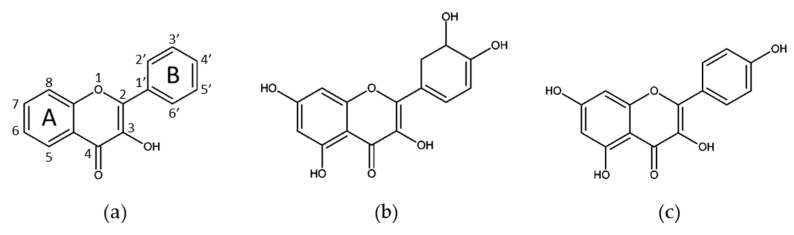
(**a**) Basic structure of flavonols. (**b**) Kaempferol. (**c**) Quercetin.

**Figure 3 molecules-25-03531-f003:**
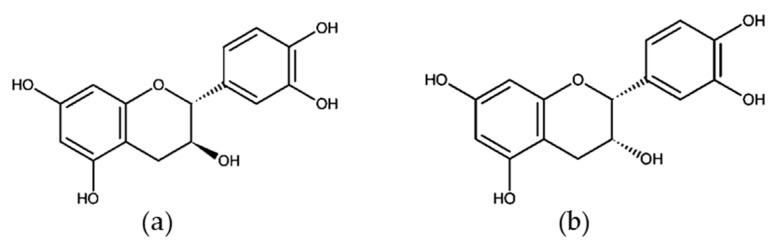
Structures of (**a**) (+)-catechin and (**b**) (−)-epicatechin.

**Figure 4 molecules-25-03531-f004:**
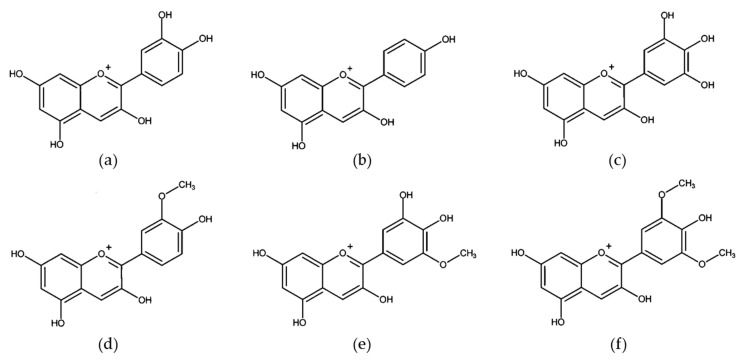
The most frequent anthocyanidins. (**a**) cyanidin, (**b**) pelargonidin, (**c**) delphinidin, (**d**) peonidin, (**e**) petunidin, (**f**) malvidin.

**Figure 5 molecules-25-03531-f005:**
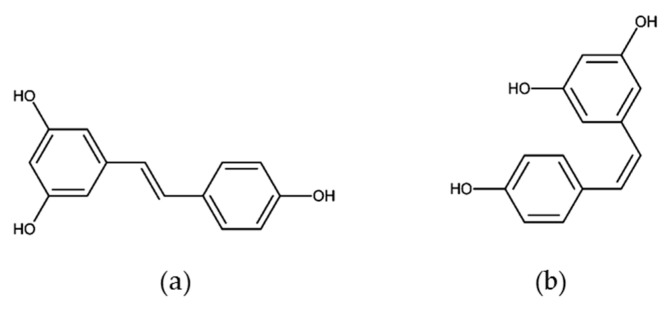
Structures of stilbenes. (**a**) Trans- resveratrol and (**b**) cis-resveratrol.

**Table 1 molecules-25-03531-t001:** Summary of metabolites found or used in different in vitro and in vivo models.

Dietary Factor/Isolated Compound	Model (Human/Animal/Cancer Cell Lines)	Sample Type	Discriminating Metabolites	Average of Metabolites (Found or Used)	Primary Reference
Red wine	Human (healthy subjects)	Plasma	catechin gallic acid 4-*O*-methylgallic acid 3-*O*-methylgallic acid caffeic acid	0.13–1.5 µmol/L	[197]
Red wine resveratrol in capsules	Human (healthy subjects)	Plasma	*trans*-resveratrol *trans*-resveratrol-4′-*O*-glucuronide *trans*-resveratrol-3′-*O*-glucuronide resveratrol resveratrol-*O*-glucuronide resveratrol-*O*-sulfate	0–0.03 µM 0–3.9 µM 0–2.7 µM 0.04–0.23 µM 0.56–2.90 µM 0.75–4.78 µM	[201]
Red wine and red grape juice	Human (male healthy subjects)	Plasma Urine	cyanidin 3-glucoside, delphinidin 3-glucoside, malvidin 3-glucoside, peonidin 3-glucoside, petunidin 3-glucoside	0.42–48.8 ng/mL (maximum) 0.66–86.7 µg/h	[218]
Red wine, dealcoholized red wine and grape juice	Human (male healthy subjects)	Plasma Urine	malvidin-3-glucoside	1.38 nM (maximum) 13.3–27.0 µg	[217]
Habitual diets	Human (healthy subjects)	Urine	3,4-dihydroxyphenylacetic acid *m*-hydroxyphenylacetic acid homovanillic acid	0.7 µg/mL 4.8 µg/mL 2.8 µg/mL	[225]
Red wine and dealcoholized red wine	Human (healthy subjects)	Urine	catechin (unmethylated conjugates) catechin (methylated conjugates)	5.32 µmol 1.27 µmol	[210]
Wine	Human (healthy subjects)	Urine	gallic acid 4-*O*-methylgallic acid	1.6–6.1 µmol/d	[198]
Red wine and dealcoholized red wine	Human (healthy adults)	Human feces	3,5-dihydroxybenzoic acid, protocatechuic acid, 3-*O*-methylgallic acid, vanillic acid, syringic acid, p-coumaric acid, phenylpropionic acid, 4-hydroxy-5-(phenyl)valeric acid, 2-hydroxyglutaric acid, 2-methylbutyric acid, 2,3-pentanedione, diethylmalonate, 2-phenethyl butyrate, 2-phenylethyl hexanoate, 5-(3′,4′-dihydroxyphenyl)gamma-valerolactone, 3-(3′-hydroxyphenyl)propionic acid, 4-hydroxy-5-(3′-hydroxyphenyl)valeric acid, benzoic acid, 4-hydroxy-5-(phenyl)valeric acid	0.2–50 µg/g	[191]
Catechin	Human (healthy subjects)	Feces	4-hydroxybenzoic acid 2,4,6-trihydroxybenzoic acid phloroglucinol 4-methoxysalicylic acid	Not determined	[211]
*Trans*-resveratrol	Human (healthy subjects)	Feces	dihydroresveratrol 3,4′-dihydroxy-*trans*-stilbene 3,4′-dihydroxybibenzyl (lunularin)	0–86.9 µmol/L 0–11.1 µmol/L 0–79.8 µmol/L	[203]
Fried onions, quercetin rutinoside, quercetin aglycone	Human (healthy ileostomy subjects)	Ileostomy effluent urine	quercetin	37–72 mg 73–275 µg	[223]
Isotopically labeled cyanidin-3-glucoside (6,8,10,3′,5′-13C_5_-C3G)	Human (male healthy subjects)	Serum Urine	24 labeled metabolites were identified (cyanidin-glucuronide, methyl cyanidin-glucuronide, methyl C3G-glucuronide, protocatechuic acid (PCA), phloroglucinaldehyde, phloroglucinaldehyde, PCA-3-glucuronide, PCA-4-glucuronide, PCA-3-sulfate, PCA-4-sulfate, vanillic acid, isovanillic acid, vanillic acid-glucuronide, isovanillic acid-glucuronide, vanillic acid-sulfate, isovanillic acid-sulfate, methyl 3,4-dihydroxybenzoate, 2-hydroxy-4-methoxybenzoic acid, methyl vanillate, 3,4-dihydroxyphenylacetic acid, 4-hydroxyphenylacetic acid, caffeic acid, ferulic acid, hippuric acid)	6.11 µmol/L (maximum) 15.69 µmol/L (maximum)	[220]
Red wine powder	Animal (male Wistar rats)	Urine Plasma	aromatic acids catechins hippuric acid hippuric acid	4.7–2790 µg/d 0–8 mg/d 0.6–3 mg/d 60–110 µmol/L	[193]
(+)-Catechin	Animal (male Wistar rats)	Plasma	catechin glucuronide catechin glucuronide + sulfate 3′-*O*-methyl catechin-glucuronide 3′-*O*-methyl catechin-glucuronide + sulfate	0.2–2.8 µmol/L 0.1–0.8 µmol/L 0.3–19.3 µmol/L 16.8–38.3 µmol/L	[208]
(+)-Catechin (−)-Epicatechin (+)-Catechin + (−)-Epicatechin	Animal (male Sprague-Dawley rats)	Plasma Urine	catechin epicatechin 3′-*O*-methyl-catechin 3′-*O*-methyl-epicatechin catechin epicatechin 3′-*O*-methyl-catechin 3′-*O*-methyl-epicatechin	0.15–44.2 µmol.h.L^−1^ 0–41.9 µmol.h.L^−1^ 0–23.0 µmol.h.L^−1^ 0.82–78.3 µmol.h.L^−1^ 0.01–8.85 µmol.h.L^−1^ 0.03–16.6 µmol.h.L^−1^ 0–3.60 µmol.h.L^−1^ 0–9.45 µmol.h.L^−1^	[209]
Cyanidin 3-*O*-β-D-glucoside	Animal (male Wistar rats)	Plasma Kidney Liver	cyanidin 3-*O*-β-d-glucoside protocatechuic acid cyanidin 3-*O*-β-d-glucoside methylated cyanidin 3-*O*-β-d-glucoside cyanidin 3-*O*-β-d-glucoside methylated cyanidin 3-*O*-β-d-glucoside	0–0.31 µmol/L 0–2.56 µmol/L 0–3.20 µmol/L 0–1.32 µmol/L 0 µmol/L 0–0.64 µmol/L	[215]
Rutin Quercetin	Animal (rabbits)	Urine	3,4-dihydroxyphenylacetic acid *m*-hydroxyphenylacetic acid *p*-hydroxyphenylacetic acid homovanillic acid	Not determined	[224]
Phloroglucinol	Animal (athymic Balb/c female nude mice)	Mice	phloroglucinol	25 mg of phloroglucinol/kg of body	[213]
Hippuric acid associated with doxorubicin or oxaliplatin	Cancer cell lines (MDA-MB-231, MCF-7, Caco-2)	Cells	Hippuric acid associated with doxorubicin or oxaliplatin	0.13–20 µg/mL (IC_50_)	[194]
4-hydroxybenzoic acid	Cancer cell lines (MCF-7, adriamycin-resistant cells MCF-7/ADM, MDA-MB-231, MDA-MB-468, 4T1) Animal (BALB/c mice)	Cells Tumor	4-hydroxybenzoic acid	0–20 µM 2 mg/Kg	[195]
Protocatechuic acid	Cancer cell lines (MCF-7, A549, HepG2, HeLa, LNCap)	Cells	protocatechuic acid	1–8 µmol/L	[196]
Gallic acid	Cancer cell lines (MDA-MB-231, HS578T, MCF-7)	Cells	gallic acid	5–400 µM	[199]
Resveratrol Hydrosystilbenes Dihydroresveratrol	Cancer cell lines (MCF-7, MDA-MB-231, BT-474, K-562)	Cells	resveratrol hydrosystilbenes dihydroresveratrol	1 nM–10 µM	[206]
Resveratrol-3-*O*-sulfate	Cancer cell line (MCF-7)	Cells	resveratrol-3-*O*-sulfate	500 nM–100 µM	[205]
Resveratrol Resveratrol-3′-*O*-glucuronide Resveratrol 3′-*O*-sulfate Resveratrol 4′-*O*-sulfate Dihydroresveratrol Dihydroresveratrol-3′-*O*-glucuronide	Cancer cell lines (MCF-7, MDA-MB-231)	Cells	resveratrol resveratrol-3′-*O*-glucuronide resveratrol 3′-*O*-sulfate resveratrol 4′-*O*-sulfate dihydroresveratrol dihydroresveratrol-3′-*O*-glucuronide	0.4–10 µmol/L	[207]
Phloroglucinol	Cancer cell lines (BT549, MDA-MB-231, MCF-7, SK-BR3, BT549)	Cells	phloroglucinol	0–100 µM	[212]
Delphinidin-3-glucuronide Cyanidin-3-glucuronide Petunidin-3-glucuronide	Cancer cell lines (MKN-28, Caco-2, MCF-7)	Cells	delphinidin-3-glucuronide cyanidin-3-glucuronide petunidin-3-glucuronide	6.3–100 µM	[221]
Quercetin-3-*O*-glucuronide	Non-tumorigenic cell line (MCF-10A) and cancer cell line (MDA-MB-231)	Cells	quercetin-3-*O*-glucuronide	0.01–µM	[226]

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
