# Peer review of "Bioactive Compounds and Metabolites from Grapes and Red Wine in Breast Cancer Chemoprevention and Therapy"

_molecules, 2020, doi:10.3390/molecules25153531_

Round 1

Reviewer 1 Report

This is a very interesting a well-written review on the effects of compounds, mostly polyphenols, from grapes and red wine on breast cancer development and prognosis.

Some major comments:

  1. The review is quite long. The authors give too much detail on bioavailability and food sources of the polyphenols.
  2. Why did the authors separate the results from grape and red wine? The compounds present in both dietary sources are similar.
  3. A lot of the polyphenols included in the review are not characteristics of the grapes or red wine, such as catechins that are much more abundant in tea (for example)
  4. There is a lot of great information on in vitro studies, less in animal models, and almost nothing in human studies. This issue needs to be solved. Please add more data on clinical and epidemiological studies.
  5. Red wine contains alcohol, a well-known risk factor for breast cancer, so is it worth it to include wine in this review? Or it is better just to talk about grape polyphenols, which are “the same” as in red wine, but you do not have the problem of alcohol. The alcohol issue is a very important Public Health concern.

Minor comments:

  1. Line 41-43, which epidemiological studies are you referring to? Please add references. It is strange to find epidemiological studies that showed a reduced risk between wine consumption and breast cancer risk, since alcohol is a well-known risk factor of breast cancer.
  2. Line 50-51, why do you include anthocyanins and anthocyanidins as different subclasses of flavonoids? The only difference is that one is the aglycone and the other is the glycoside.
  3. Line 77 and 224, please use the abbreviation GSP
  4. Line 110-115. Please delete this paragraph because it is not related to grapes/wine or breast cancer.
  5. Line 234-235, failed “to” decrease
  6. Line 237, keep only the abbreviation GSPE

Author Response

Reviewer 1

Comments and Suggestions for Authors

This is a very interesting a well-written review on the effects of compounds, mostly polyphenols, from grapes and red wine on breast cancer development and prognosis.

Some major comments:

1. The review is quite long. The authors give too much detail on bioavailability and food sources of the polyphenols.

Answer: In this review, we explore a wide range of available studies regarding the role of bioactive compounds from grapes and red wine and their metabolites on breast cancer, to provide an update on what has already been done and can be found in the literature. We believe that detailed information on bioavailability and food sources of phenolic compounds can be quite useful for non-specialist researchers in the field.

2. Why did the authors separate the results from grape and red wine? The compounds present in both dietary sources are similar.

Answer: Although the phenolic compounds present in grapes and red wine are similar, we consider the differences in their concentration, structure and bioavailability, which may vary depending on the food source, particularly because of the processing of grapes, wine production and storage.

3. A lot of the polyphenols included in the review are not characteristics of the grapes or red wine, such as catechins that are much more abundant in tea (for example).

Answer: In order to explore several individual grape seed components, we also considered low molecular weight polyphenolic constituents of grape seed as catechin and epicatechin for this review. Although these flavonoids are major components of green tea, they can also be found in grape seeds and gallic acid, procyanidins and epigallo-catechins overall accounting for about 80-90% of dry extract (Guendez et al, 2005. Food Chemistry, 89:1–9). We believe that contribution of every bioactive compound, even the less represented molecules cannot be excluded, given that some biological functions seem to be synergistically afforded by interactions among the different components (Pignatelli et al, 2000. Am J Clin Nutr,72:1150–5).

4. There is a lot of great information on in vitro studies, less in animal models, and almost nothing in human studies. This issue needs to be solved. Please add more data on clinical and epidemiological studies.

Answer: To accomplish this suggestion the following sentences were included in the text:

  • Lines 97-101: “Grape seed proanthocyanidin extract (GSPE) has also a promising therapeutic role against adverse effects of chemotherapeutic agents like carboplatin and thalidomide. GSPE acts as a potent natural antioxidant and exhibited a protective effect on liver and heart tissues against carboplatin and thalidomide -induced damage in rats”.

  • Lines 470-472: “In 2005, it was shown for the first time that resveratrol from grape consumption is inversely related to breast cancer risk, as reported in a case-control study conducted between 1993 and 2003 in the Swiss Canton of Vaud on 369 cases and 602 controls”.

5. Red wine contains alcohol, a well-known risk factor for breast cancer, so is it worth it to include wine in this review? Or it is better just to talk about grape polyphenols, which are “the same” as in red wine, but you do not have the problem of alcohol. The alcohol issue is a very important Public Health concern.

Answer: We agree that alcoholic drinks of any kind raise breast cancer risk and, for this reason, we did not address studies that link red wine consumption with cancer prevention in this review. We have included studies that discuss the effects of isolated bioactive compounds present in red wine, especially because of the role of alcohol in converting these compounds into bioactive metabolites, as described in section 5.

Minor comments:

1. Line 41-43, which epidemiological studies are you referring to? Please add references. It is strange to find epidemiological studies that showed a reduced risk between wine consumption and breast cancer risk, since alcohol is a well-known risk factor of breast cancer.

Answer: We consider the commentary and the sentence was rewritten, with a new reference included: “Several in vitro and in vivo studies have reported the beneficial effects promoted by bioactive compounds from grapes and winery by-products” (lines 40-41).

2. Line 50-51, why do you include anthocyanins and anthocyanidins as different subclasses of flavonoids? The only difference is that one is the aglycone and the other is the glycoside.

Answer: In fact, we did not consider separating them into different subclasses of flavonoids. In section 3.3, we described both anthocyanins (glycosides) and anthocyanidins (aglycones), because we address the antitumor potential of total anthocyanin-enriched grape extracts, as well as the effects of isolated anthocyanidins against breast cancer.

3. Line 77 and 224, please use the abbreviation GSP.

Answer: We consider the commentary and the requested correction was made (lines 89 and 255).

4. Line 110-115. Please delete this paragraph because it is not related to grapes/wine or breast cancer.

Answer: We considered the suggestion and this paragraph was removed from the text.

5. Line 234-235, failed “to” decrease.

Answer: We consider the commentary and the requested correction was made (line 267).

6. Line 237, keep only the abbreviation GSPE.

Answer: We consider the commentary and the requested correction was made (line 270).

Reviewer 2 Report

The review “Bioactive compounds and metabolites from grapes and red wine in breast cancer chemoprevention and therapy” for publication in Molecules examines the current studies about the topic, giving evidence of the bioactive compounds and metabolites found in grapes and red wine, and their effects in vitro and in vivo

Summarizing the information about the effect of consuming grapes and red wine related to breast cancer is indispensable to understand the “state of art” of this theme and opening new research line in the topic.

Before publication, major revisions are necessary to improve the coherence and uniformity of the text:

  • Abstract: why authors described “food sources” if the present review involves only grapes and red wine?

  • English grammar should be revised throughout the text.

Indeed, there is a mix of present and past tenses (past simple and present perfect). Results from previous works are usually described in past simple. Most of the sections describing previous works are written in present tense. As examples, in the following sentences there are a mix of verbal tenses:

The resveratrol intervention (1 g 500 daily, for 12 weeks) did not result in significant changes in serum concentrations of estradiol, 501 estrone, or testosterone, but has favorable effects on estrogen metabolism and steroid 502 hormone-binding globulin (SHBG) [172].

 After rats ate cyanidin-3-glucoside, the aglycone is only found in the small intestine,…

  • Section 2 needs to be reorganized, as authors defined cancer metastasis after describing some effects of GSP in breast metastatic cancer cells (MDA-MB and MCF7).

  • Line 54: What is containing this polyphenolic fraction (3)? Please, add information about the identification and quantification of phenolic compounds, and/or type of extraction. This fact could help other authors to understand the group of polyphenols involved in the beneficial effect in the cells. The same regarding Kijima et al., (7) results, reference (84),

  • Please, add a little explanation about the kind of breast cancer cells, which are more feasible? Useful? More reported?

  • Paragraph lines 54-57: Use the past tense, according to the rest of the paragraph. The same thorough the text, as examples: lines 80, 130, 167,

  • Please, rewrite this sentence for a better understanding:

“Changes in the functionality of connexin proteins - structural proteins of the channel-forming gap junctions - modulate cell-cell communication via gap-junctional intercellular communication, which have been shown to be involved in the apoptotic process [4].”

  • Use µg instead of ug.

  • Section 2: In my opinion, it could be interesting to describe which beneficial effects related to GSP and GSE have been widely reported, concluding a feasible effect of seed grapes.

  • Lines 110-113: In my opinion this information is not relevant in this work:

 “Quercetin (2-(3,4-Dihydroxyphenyl)-3,5,7-trihydroxy-4H-1-benzopyran-4-one), is most abundantly found in vegetables such as onions, berries and apples, in which it is produced as a secondary metabolite. Kaempferol (3,5,7-Trihydroxy-2-(4-hydroxyphenyl)-4H-1-benzopyran-4-one) was first encountered in tea (Camellia sinensis), but is present in different vegetables and herbs, such as grapes, onions, capers, tomatoes and broccoli.””

  • Please, rewrite this sentence for a better understanding:

“The same was observed for kaempferol cells in a 3-D model, 132 with ERK signaling being responsible for the apoptotic death.”

  • Line 130: breast cancer cells (30–32). Please, add examples of the names of these cell lines between parentheses.

  • Line 141: breast cancer cells (39-41): Please, add examples of the names of these cell lines between parentheses.

  • Line 163: What is the meaning of 10-5 M range? 10 µM? Please use similar units for the revised results.

  • Line 179: Which compounds are “these compounds”?

  • Authors should cite studies about grape or red wine (poly)phenols bioavailability instead of tea bioactives bioavailability? If possible. (lines 195-199).

  • What is new in Lines 228-229 regarding the study of Kijima et al., compared to lines 77-78?

  • Authors compare references (93) and (94), please, add the doses of GS proanthocyanidins used in these works.

  • Section 3.3. is very well written, it contains more detailed information compared to the previous one, for example, the doses of the treatments, which are very important to estimate the health-related effects of (poly)phenols.

  • Lines 309-311: What is the difference between “inhibitory activity” and “inhibited cancer cell proliferation”? Please rewrite it for a better understanding.

  • Line 550: What is the meaning of “another metabolite”? “4-Hydroxybenzoic acid (4-HBA), another metabolite and a histone deacetylase 6 (HDAC6) inhibitor,… ”

  • Section 5. Generally, please add detailed information about doses of bioactive metabolites that caused a positive effect against cancer cells.

  • Line 603-604: Please, explain it better: “A significant increase in MCF-7 cancer cells growth rates was 603 shown in the presence of picomolar concentrations of dihydroresveratrol (DH-RSV).”

  • Lines 662-663: How much was enhanced the Cy-3-gls metabolism? Did this publication (206) suggest a positive effect of ethanol? Please, discuss it.

  • Regarding reference 208, it would be interesting that anthocyanin-derived metabolites showed similar antiproliferative effect than anthocyanins. Please, discuss it better.

  • Line 693: Authors may link these two sentences: There was evidence that metabolic changes were correlated with metastasis and metabolism of tumors [217–220]. Metabolism changes are often associated with resistance to chemotherapy and therapeutic sensitivity in clinical chemotherapy.

  • Line 704: Please, add here a wide range of bioactives or metabolites that may cause anticancer activity against breast cancer, in agreement with the works revised in the present review

Conclusions regarding the title of the work should be stated. Please, enrich this section.

  • In my opinion, the following information from the Conclusions belongs to Results, as well as the Figure 5.

 The major anticancer activities promoted by these compounds are summarized in figure 5 and includes modulation of estrogen cell signaling, cancer cell differentiation, cell growth inhibition, apoptosis induction and suppression of the metastatic behavior.

  • Please, rewrite this sentence for a better understanding:

Bioavailability of these compounds brings another important issue that must be circumvented by 718 nanotechnology to improve their local biological effects

  • Is the following sentence a conclusion of the work? There are still not clinical trials showing this conclusion.

In this way, the polyphenols present in grape and wine have long been used as sources of lead compounds for breast cancer chemotherapy improvement…

  • Improve the meaning of this sentence: “along with the technology improvements necessary for their application”

Author Response

Reviewer 2

Comments and Suggestions for Authors

The review “Bioactive compounds and metabolites from grapes and red wine in breast cancer chemoprevention and therapy” for publication in Molecules examines the current studies about the topic, giving evidence of the bioactive compounds and metabolites found in grapes and red wine, and their effects in vitro and in vivo

Summarizing the information about the effect of consuming grapes and red wine related to breast cancer is indispensable to understand the “state of art” of this theme and opening new research line in the topic.

Before publication, major revisions are necessary to improve the coherence and uniformity of the text:

  • Abstract: why authors described “food sources” if the present review involves only grapes and red wine?

Answer: We prefer to remove “food sources” and the sentence was modified to: “Flavonoid compounds like flavonols, monomeric catechins, proanthocyanidins, anthocyanins, anthocyanidins and non-flavonoid phenolic compounds, such as resveratrol, as well as their metabolites, are discussed with respect to structure and metabolism/bioavailability” (lines 29-32).

  • English grammar should be revised throughout the text. Indeed, there is a mix of present and past tenses (past simple and present perfect). Results from previous works are usually described in past simple. Most of the sections describing previous works are written in present tense. As examples, in the following sentences there are a mix of verbal tenses:

The resveratrol intervention (1 g 500 daily, for 12 weeks) did not result in significant changes in serum concentrations of estradiol, 501 estrone, or testosterone, but has favorable effects on estrogen metabolism and steroid 502 hormone-binding globulin (SHBG) [172].

 After rats ate cyanidin-3-glucoside, the aglycone is only found in the small intestine,… 

Answer: The revised version of the paper includes several editions, distributed along the text, for elucidation and English improvement and are highlighted with the Microsoft Word “Track Changes” function in the file.

  • Section 2 needs to be reorganized, as authors defined cancer metastasis after describing some effects of GSP in breast metastatic cancer cells (MDA-MB and MCF7).

Answer: For better understanding of the studies presented, section 2 was reorganized as requested. All editions are highlighted with the Microsoft Word “Track Changes” function in the file.

  • Line 54: What is containing this polyphenolic fraction (3)? Please, add information about the identification and quantification of phenolic compounds, and/or type of extraction. This fact could help other authors to understand the group of polyphenols involved in the beneficial effect in the cells. The same regarding Kijima et al., (7) results, reference (84).

Answer: The phenolic fraction and extracts obtained from grape seed are mainly sources of procyanidins and catechins. For better understanding of polyphenols involved in the beneficial effect, we included all polyphenols content available on the originals articles before each extract.

  • Please, add a little explanation about the kind of breast cancer cells, which are more feasible? Useful? More reported?

Answer: A paragraph in the end of the “introduction” section was added, with the information requested by the referee about cell line models of breast cancer.

  • Paragraph lines 54-57: Use the past tense, according to the rest of the paragraph. The same thorough the text, as examples: lines 80, 130, 167.

Answer: Verb tenses were revised throughout the text.

  • Please, rewrite this sentence for a better understanding:

“Changes in the functionality of connexin proteins - structural proteins of the channel-forming gap junctions - modulate cell-cell communication via gap-junctional intercellular communication, which have been shown to be involved in the apoptotic process [4].” 

Answer: The sentence was modified to: “Polyphenols obtained by hydroalcoholic extraction from grape seeds promoted a selective inhibition of cell viability and induction of apoptotic cell death on MCF-7 cells. The authors hypothesize that this effect is mediated by gap-junction-mediated cell-cell communications improvement via re-localization of Cx43 proteins and up-regulation of CX43 gene, since gap junctions have been associated to the apoptotic process” (lines 69-73).

  • Use µg instead of ug

Answer: We considered the commentary and the requested correction was made throughout the text.

  • Section 2: In my opinion, it could be interesting to describe which beneficial effects related to GSP and GSE have been widely reported, concluding a feasible effect of seed grapes.

Answer: We included the sentences (lines 250-255): GSE and GSPE have attracted attention because of their health effects. They improve glucose homeostasis and antioxidant cell defenses, reduce platelet aggregation, adhesion, and generation of superoxide anion; prevent the increase of low-density lipoprotein (LDL) cholesterol concentration; reduce plasma C-reactive protein; decreases the expression of the proinflammatory cytokine tumor necrosis factor alpha (TNF-α) and interleukin 6 (IL-6) and also inhibit Gram-positive and Gram-negative bacteria”.

  • Lines 110-113: In my opinion this information is not relevant in this work:

 “Quercetin (2-(3,4-Dihydroxyphenyl)-3,5,7-trihydroxy-4H-1-benzopyran-4-one), is most abundantly found in vegetables such as onions, berries and apples, in which it is produced as a secondary metabolite. Kaempferol (3,5,7-Trihydroxy-2-(4-hydroxyphenyl)-4H-1-benzopyran-4-one) was first encountered in tea (Camellia sinensis), but is present in different vegetables and herbs, such as grapes, onions, capers, tomatoes and broccoli.” 

Answer: We considered the suggestion and the paragraph was removed from the text.

  • Please, rewrite this sentence for a better understanding:

“The same was observed for kaempferol cells in a 3-D model, 132 with ERK signaling being responsible for the apoptotic death.” 

Answer: The sentence was improved to: “The same was observed for kaempferol in the MCF-7 3-D model, with ERK signaling...” (lines 148-149).

  • Line 130: breast cancer cells (30–32). Please, add examples of the names of these cell lines between parentheses.

Answer:  The examples of cell lines were included, as suggested by the referee.

  • Line 141: breast cancer cells (39-41): Please, add examples of the names of these cell lines between parentheses.

Answer: The examples of cell lines were included, as suggested (lines 157-158).

  • Line 163: What is the meaning of 10-5M range? 10 µM? Please use similar units for the revised results.

Answer: In order to improve comprehension, we have modified the concentration units in this sentence to the same found throughout the text (µM).

  • Line 179: Which compounds are “these compounds”? 

Answer: The sentence refers to both quercetin and kaempferol, thus, it was modified to: “Multidrug transporters are also acknowledged in the metabolization/elimination of quercetin and kaempferol” (lines 200-202).

  • Authors should cite studies about grape or red wine (poly)phenols bioavailability instead of tea bioactives bioavailability? If possible. (lines 195-199).

Answer: We included the sentences (lines 222-226): “Bioavailability of procyanidins closely resembles that of flavan-3-ol monomers. Different studies, following the ingestion of GSE and GSPE have shown that during digestion, the oligomers are fragmented into monomeric units of (+)-catechin and (-)-epicatechin and free forms of dimers and trimers have been detected in rat plasma [97,98]. Procyanidin B1 was also detected in human serum 2 h after intake of GSE [99]”.

  • What is new in Lines 228-229 regarding the study of Kijima et al., compared to lines 77-78?

Answer: Kijima et al. have examined the effects of GSE on aromatase activity and expression by using in vitro experiments (lines 89-90). However, the in vivo results with mice showed a tumor reduction promoted by GSE, but this finding is not limited to the suppression of aromatase activity (lines 260-262).

  • Authors compare references (93) and (94), please, add the doses of GS proanthocyanidins used in these works. 

Answer: We included the GSPE doses (line 276).

  • Section 3.3. is very well written, it contains more detailed information compared to the previous one, for example, the doses of the treatments, which are very important to estimate the health-related effects of (poly)phenols. 

Answer: We consider the commentary and section 3.2 was reformulated, with the insertion of more detailed information about the content of phenolic compounds present in the extracts used and treatments, as described before.

  • Lines 309-311: What is the difference between “inhibitory activity” and “inhibited cancer cell proliferation”? Please rewrite it for a better understanding.

Answer: We consider the commentary and the sentence was rewritten: “Although anthocyanins did not inhibit proliferation of any cell line tested” (lines 341-342).

  • Line 550: What is the meaning of “another metabolite”? “4-Hydroxybenzoic acid (4-HBA), another metabolite and a histone deacetylase 6 (HDAC6) inhibitor,… ”

Answer: It means that 4-Hydroxybenzoic acid (4-HBA) is another metabolite studied, because we started to describe it in the sentence. Based on the comment, we prefer to remove "another metabolite" from the sentence. Line 580: “4-Hydroxybenzoic acid (4-HBA) and a histone deacetylase 6 (HDAC6) inhibitor…”

  • Section 5. Generally, please add detailed information about doses of bioactive metabolites that caused a positive effect against cancer cells.

Answer: We summarize the data obtained in section 5: Wine metabolites and breast cancer (in vitro and in vivo studies), which refers to the metabolites of wine or grape, in Table 1. We suggest the inclusion, not only with cancer cells, but also with all models utilized in this review at the end of this section, the following text (Lines 754-755): Table 1 summarizes the data collected from the literature about the metabolites dosage used or found in the different in vitro and in vivo models mentioned in this review.

  • Line 603-604: Please, explain it better: “A significant increase in MCF-7 cancer cells growth rates was 603 shown in the presence of picomolar concentrations of dihydroresveratrol (DH-RSV).”

Answer: In hormone-sensitive tumor cell line, such as MCF-7 cells, very low concentrations of dihydroresveratrol may indicate a hormone-like effect for the compound. In other cancer cell lines (MDA-MB-231, BT-474 and K-562) that do not express hormone receptors, the proliferative effect of dihydroresveratrol was not observed. Based on comment, we changed the phrase (lines 632-635): A significant increase in MCF-7 cancer cells growth rates was shown in the presence of picomolar concentrations of dihydroresveratrol (DH-RSV),because this polyphenol has a profound proliferative effect on hormone-sensitive tumor cell lines such as MCF-7.

  • Lines 662-663: How much was enhanced the Cy-3-gls metabolism? Did this publication (206) suggest a positive effect of ethanol? Please, discuss it.

Answer: The sentence was modified (lines 693-697): Ethanol enhances cyanidin-3-O-glucoside (Cy3glc) metabolism potentiating its conversion into methylated and glucuronidated derivatives, showing an increase in the two main anthocyanin conjugates, Me-Cy-glucr and 3`-Me-Cy3glc, of 59 and 57%, respectively. In this case, the food matrix used was blackberry puree with or without ethanol, and not wines or grapes [219].

The publication of Marques et al. (2016) suggest that ethanol could accelerate the conversion of anthocyanins into methylated derivatives; but this effect was more pronounced in the overweight/obese group, in whom Cy3glc metabolism appeared to be compromised. It seems that ethanol is more effective in promoting Cy3glc metabolism when its metabolism is impaired. The authors concluded that the kinetic of these compounds is influenced by ethanol and body composition. Other important observation is that the food source utilized in this study was blackberry puree and not grape or wine and was utilized only to discuss this theme.

  • Regarding reference 208, it would be interesting that anthocyanin-derived metabolites showed similar antiproliferative effect than anthocyanins. Please, discuss it better. 

Answer: These sentences were included (lines 708-717): The methylation reaction alters the number of hydroxyl and methoxyl groups in ring B, so these metabolites are likely to have different antioxidant activities in comparison with the original anthocyanins. Generally, the health effects of anthocyanins are associated with an increase in the endogenous antioxidant defenses. In a paper by Fernandes et al. (2013) [221] the synthetized metabolites still displayed some antiproliferative activity for the three cell lines although not so intense as genuine anthocyanin. The biological studies conducted with the metabolites in comparison with the native compounds allow understanding of the real contribution of methylation towards the antioxidant and antiproliferative effects of anthocyanins. However, this subject is new and needs more publications for a good discussion, especially from methylated anthocyanin-derived metabolites.

  • Line 693: Authors may link these two sentences: There was evidence that metabolic changes were correlated with metastasis and metabolism of tumors [217–220]. Metabolism changes are often associated with resistance to chemotherapy and therapeutic sensitivity in clinical chemotherapy.

Answer: We included the sentences (lines 739-743): Cancer and metabolism are deeply interconnected, studies indicate that cancer evolution is associated with abnormal glucose metabolism that is related to high proliferation, metastasis and clinical characteristics and is allied to the action of a particular drug. In this context, chemoresistance enables cancer cells to survive drug action and proliferate uncontrollably, which may lead to strong metastatic potential and cancer progression [230–234].

  • Line 704: Please, add here a wide range of bioactives or metabolites that may cause anticancer activity against breast cancer, in agreement with the works revised in the present review.

Answer: We believe that the inclusion of Table 1 is able to answer this question.

Conclusions regarding the title of the work should be stated. Please, enrich this section.

Answer: We included these sentences in the conclusion (lines 778 – 784): “Based on dietary factor (wine or grape) or their metabolites used in different in vitro and in vivo studies for breast cancer, we conclude that there is a great variation of doses utilized or found. When the studies utilize wine or grape as bioactive compound source, it is possible to observe a great variation on metabolites quality and quantity. On the other hand, when the isolated metabolite or it its precursor were used, mainly in cancer cell lines, variations between nM until 100 µM were used, and some authors justify the use of these concentrations to approximate the physiological concentrations”. Additionally, we change a few words in the last paragraph.

  • In my opinion, the following information from the Conclusions belongs to Results, as well as the Figure 5.

 The major anticancer activities promoted by these compounds are summarized in figure 5 and includes modulation of estrogen cell signaling, cancer cell differentiation, cell growth inhibition, apoptosis induction and suppression of the metastatic behavior. 

Answer: Since this Review article does not have a “results” section, Figure 5 was prepared to summarize the anticancer activities promoted by phenolic compounds and their metabolites from grapes and red wine in the “conclusion” section.

  • Please, rewrite this sentence for a better understanding:

Bioavailability of these compounds brings another important issue that must be circumvented by 718 nanotechnology to improve their local biological effects 

Answer: The sentence was rewritten (lines 785-789) and modified to: Bioavailability of these compounds is another important issue that must be circumvented by nanotechnology to improve local biological effects. In this way, the grape and wine have long been used as sources of lead compounds in the search for breast cancer chemotherapy candidates and should be further explored in clinical studies, along with the biotechnological improvements necessary for their application.

  • Is the following sentence a conclusion of the work? There are still not clinical trials showing this conclusion.

In this way, the polyphenols present in grape and wine have long been used as sources of lead compounds for breast cancer chemotherapy improvement…

Answer: As answered in the previous question, we change the last sentence of the conclusion to: “In this way, the grape and wine have long been used as sources of lead compounds in the search for breast cancer chemotherapy candidates and should be further explored in clinical studies, along with the biotechnological improvements necessary for their application” (lines 786-789).

  • Improve the meaning of this sentence:“along with the technology improvements necessary for their application”

Answer: More adequate terms were added to improve the understanding of this sentence.

Other minor modifications proposed by the authors are indicated throughout the text with the Microsoft Word “Track Changes” function in the file.

Round 2

Reviewer 2 Report

The current version of the review has been improved, however, still some major corrections are necessary for publication. Please, find hereafter the revisions:

  • Line 32, Abstract: “end” use no italics

  • By-products from the wine industry are only appointed in line 42. This does not make sense if authors do not describe the beneficial effects of bioactive compounds derived from by-products thorough the text. Please, add some information about wine by-products in the different sections, where applicable.

  • Lines 59 and 60:

“Studies using grape seed compounds have identified various molecular targets involved in breast cancer cell differentiation, cell cycle arrest, apoptosis and metastasis, a multistep process involving migration, adhesion and invasion of cancer cells”.

What authors mean with “Studies using grape seed compounds have identified various molecular targets…”? I understand that GS compounds have a beneficial effect by their action on molecular mechanisms involved in breast cancer cell differentiation, cell cycle arrest, apoptosis and metastasis. Please, rewrite this sentence to facilitate the comprehension by the reader.

Why authors include “metastasis” in the following sentence? Are grape seed compounds able to inhibit metastasis by their action in cell cycle arrest and apoptosis? Please, rewrite this sentence to facilitate the comprehension by the reader.

  • Lines 85-90, Section 2: Verbal tenses should be revised and corrected in the last paragraph of Section 2. Past verbal tenses should be used to describe the results from previous works.

  • Please, rewrite this sentence for a better understanding:

“Additionally, studies proposed a strategy using GSE to reduce the high toxicity caused by chemotherapeutic agents to healthy tissues and drug resistance during chemotherapy of breast cancer. A combination of GSE (95% procyanidins) and doxorubicin seemed to synergize the drug effects in human breast cancer cells, enhancing MCF-7 cells in G1 phase and causing a strong apoptotic death in MDA-MB-468 and MDA-MB-231 cells”

  • Please, link the following two sentences and add examples about the protective effect of GSPE against side effects of chemotherapy:

“Grape seed proanthocyanidin extract (GSPE) showed also a promising therapeutic role against adverse effects of chemotherapeutic agents like carboplatin and thalidomide. GSPE acts as a potent natural antioxidant and exhibited a protective effect on liver and heart tissues against carboplatin and thalidomide induced damage in rats [12].

  • Redundant information is found in the following paragraph, please, rewrite and shorten it:

“The anticancer effects of kaempferol and quercetin have been described in several different cancer types, such as bladder, breast, prostate, ovarian, liver and colon. A special feature is given to breast cancer due to a superior number of published works. Epidemiological studies using flavonoids such as quercetin or kaempferol generally involve the observation of their dietary contribution to breast cancer prevention, most of the times, in combination with other compounds of dietary origin and their findings vary, ranging from null to positive [24–26].”

  • Lines 131-132: Regarding the following sentence, why did authors say before that “Epidemiological studies using flavonoids showed an effect in breast cancer” and now they say there is a limited variety of preclinical and clinical studies? What is the meaning of “limited variety”? Please, clarify it.

“It is important to mention that, in spite of the larger number of basic research papers published so far, there is a limited variety of pre-clinical and clinical studies using kaempferol or quercetin as anticancer agents”

  • Please, correct throughout the text the use of present verbal tenses instead of past verbal tenses to describe previous published results.

Examples: Line 143, 154, 157, 162, 166, 173, 197, 232, 263, etc.

  • Lines 178-179: Improve the writing of this sentence: “On the other hand, at higher concentrations (100 μM range), increments in estrogen concentrations were unable to block kaempferol effects, suggesting different pathways for activation by this compound [73].”

  • Line 216: Plasma (+)-catechin concentrations were also increased…

  • If it is possible, please, change the reference 96 about tea catechins presented in human plasma after 1-2h from ingestion by another related to wine, that will fit better in this work.

  • The paragraph from line 245 to 250 regarding other health effects related to GSE and GSPE, could be shorten and commented in other place in the text. In my opinion, it is not appropiate in the middle of the different results regarding anticancer effects of catechins. This paragraph (but shorten) could fit better in the introduction, or at the beginning or the end of the section 3.2.

  • Lines 320 – 325: Please, improve the link of this paragraph with the chemopreventive effect of delphinidin-3-glucoside by HOTAIR expression inhibition. In the current form is very difficult to understand.

  • Why the title of Section 5 is only related to wine compounds metabolites and grapes compounds metabolites are not named?

Then, authors speak about grape metabolites in lines 545, 560, 565, 590 etc.

Therefore, I think this section should content information about:

  1. Grape and wine metabolites and breast cancer (in vitro and in vivo studies)

  • Line 677: Authors use the abbreviation Cy3G for the first time. However, cyaniding-3-glucoside appears several times before. Please, write the complete name always or, use the abbreviation in parentheses after the full name the first time it appears in the text, and then, continue using the abbreviation the following times it appears.

At line 683, another abbreviation is used.

  • Line 683: Remember the use of the past tense in the text.

  • Line 701: Check references style in the text. “by Fernandes et al. (2013) [221]”

  • Line 702: some antiproliferative activities…

  • What do you mean with “genuine” anthocyanin?

  • Table 1 collects the metabolites derived from grape and wine compounds after ingestion or after in vitro fermentations. I do not understand why firstly “wine metabolites” are described, followed by a division by different compounds (resveratrol, catechins, anthocyanins and quercetin). What is the difference between the catechins described in reference 193 or 197, from 210 or 193? Why are these studies presented separately?

Please, reorder the table 1 for a better understanding. Maybe the first section “wine metabolites” is not necessary to avoid repetitions.

Conclusions should be better written, please, find here some suggestions:

  • In my opinion, Figure 5 collects the results of this revision. Therefore, this figure must be cited in SECTION 2. Even so, the sentence summarizing the major anticancer activities reported in the review, can be used in the conclusions.

  • What do you mean with “nanotechnology” in the conclusions? This is not understandable if authors do not describe before novel technologies to improve the bioavailability of phenolic compounds or the bioactive effects of these compounds. Therefore, this note should be better explained if you names “nanotechnology” only as a conclusion.

  • The sentence “It is also important to point out that the effects produced by the glycosidic forms and the aglycones might lead to different routes of absorption and/or metabolization, leading to important variations in bioavailability and global effects produced” could be linked with the following paragraph, where authors described that the dietary SOURCE (wine or grape) and their derived metabolites are responsible for the great variation of effects found, being different factors responsible of these effects, such as the doses used (from XX nM to 100 µM), the qualitative profile of the bioactive compounds presented.

  • It is very important to highlight which are the physiological concentrations used for phenolic compounds and anticancer activities research. If possible, please, add some of this information before conclusions.

Author Response

As requested, we enclose a revised version of this manuscript that constructively addresses all of the concerns and suggestions of the Reviewer. The lines indicated in the responses below refer to the final version submitted with the accepted changes.

Comments and Suggestions for Authors:

Reviewer 2

The current version of the review has been improved; however, still some major corrections are necessary for publication. Please, find hereafter the revisions:

Line 32, Abstract: “end” use no italics

Answer: We consider the commentary and the requested correction were made.

By-products from the wine industry are only appointed in line 42. This does not make sense if authors do not describe the beneficial effects of bioactive compounds derived from by-products thorough the text. Please, add some information about wine by-products in the different sections, where applicable.

Answer: We considered the commentary and the expression was replaced by: “bioactive compounds from grapes and its derivative products” (Lines 42 and 43).

Lines 59 and 60:

“Studies using grape seed compounds have identified various molecular targets involved in breast cancer cell differentiation, cell cycle arrest, apoptosis and metastasis, a multistep process involving migration, adhesion and invasion of cancer cells”.

What authors mean with “Studies using grape seed compounds have identified various molecular targets…”? I understand that GS compounds have a beneficial effect by their action on molecular mechanisms involved in breast cancer cell differentiation, cell cycle arrest, apoptosis and metastasis. Please, rewrite this sentence to facilitate the comprehension by the reader.

Answer: As suggested, for better understanding, the sentence was rewritten: “Several molecular pathways involved in breast cancer cell signaling and differentiation, cell cycle arrest, apoptosis, and metastasis can be modulated by these compounds” (Lines 60-61).

Why authors include “metastasis” in the following sentence? Are grape seed compounds able to inhibit metastasis by their action in cell cycle arrest and apoptosis? Please, rewrite this sentence to facilitate the comprehension by the reader.

Answer: As mentioned above, the sentence was rewritten: “Several molecular pathways involved in breast cancer cell signaling and differentiation, cell cycle arrest, apoptosis, and metastasis can be modulated by these compounds” (Lines 60-61).

Lines 85-90, Section 2: Verbal tenses should be revised and corrected in the last paragraph of Section 2. Past verbal tenses should be used to describe the results from previous works.

Answer: As recommended, the sentences were revised and adjusted, when necessary.

Please, rewrite this sentence for a better understanding:

“Additionally, studies proposed a strategy using GSE to reduce the high toxicity caused by chemotherapeutic agents to healthy tissues and drug resistance during chemotherapy of breast cancer. A combination of GSE (95% procyanidins) and doxorubicin seemed to synergize the drug effects in human breast cancer cells, enhancing MCF-7 cells in G1 phase and causing a strong apoptotic death in MDA-MB-468 and MDA-MB-231 cells”

Answer: As suggested, the sentence was rewritten: “Another approach includes the combination of therapeutic compounds, like doxorubicin (Dox), and phytochemicals for cancer management. GSE (95% procyanidins) increases the efficacy of Dox in human breast cancer MCF-7, MDA-MB468 and MDA-MB231 cells suggesting a strong possibility of a synergistic effect of GSE and Dox combination, independent of estrogen receptor status of cells”. (Lines 97-101)

Please, link the following two sentences and add examples about the protective effect of GSPE against side effects of chemotherapy:

“Grape seed proanthocyanidin extract (GSPE) showed also a promising therapeutic role against adverse effects of chemotherapeutic agents like carboplatin and thalidomide. GSPE acts as a potent natural antioxidant and exhibited a protective effect on liver and heart tissues against carboplatin and thalidomide induced damage in rats [12].

Answer: As suggested, the sentence was rewritten: “Grape seed proanthocyanidin extract (GSPE) also showed a promising therapeutic role against the adverse effects of the chemotherapeutic agents carboplatin and thalidomide. Administration of these agents in rats led to an enhancement in the TNF-α and IL-6 cytokine levels, which could be partially reversed by the administration of GSPE. Besides, GSPE reduced free radicals like thiobarbituric acid-reactive substances and nitric oxide and increased glutathione and antioxidant enzymes in liver and heart”. (Lines 101-107)

Redundant information is found in the following paragraph, please, rewrite and shorten it:

“The anticancer effects of kaempferol and quercetin have been described in several different cancer types, such as bladder, breast, prostate, ovarian, liver and colon. A special feature is given to breast cancer due to a superior number of published works. Epidemiological studies using flavonoids such as quercetin or kaempferol generally involve the observation of their dietary contribution to breast cancer prevention, most of the times, in combination with other compounds of dietary origin and their findings vary, ranging from null to positive [24–26].”

Answer: We thank the referee for the comment. The paragraph was shortened and adapted to make the sentence clearer. The final version is "The anticancer effects of kaempferol and quercetin have been described in several different cancer types, such as bladder, breast, prostate, ovarian, liver and colon. A special feature is given to breast cancer due to a superior number of published works. The few epidemiological studies available using flavonoids such as quercetin or kaempferol on breast cancer involve the observation of their dietary contribution. However, their findings vary from null to positive effects in cancer prevention [24–26].".

Lines 131-132: Regarding the following sentence, why did authors say before that “Epidemiological studies using flavonoids showed an effect in breast cancer” and now they say there is a limited variety of preclinical and clinical studies? What is the meaning of “limited variety”? Please, clarify it.

“It is important to mention that, in spite of the larger number of basic research papers published so far, there is a limited variety of pre-clinical and clinical studies using kaempferol or quercetin as anticancer agents”

Answer: When we mention a "limited variety" of pre-clinical and clinical studies using kaempferol or quercetin as anticancer agents", we refer to the studies in which these compounds are administered as potential anticancer drugs, in actual doses and not as components of a diet. Thus, the term "limited variety" was substituted by "small number", for a better comprehension.

Please, correct throughout the text the use of present verbal tenses instead of past verbal tenses to describe previous published results.

Examples: Line 143, 154, 157, 162, 166, 173, 197, 232, 263, etc.

Answer: As recommended, the whole text was revised and adjusted, when necessary.

Lines 178-179: Improve the writing of this sentence: “On the other hand, at higher concentrations (100 μM range), increments in estrogen concentrations were unable to block kaempferol effects, suggesting different pathways for activation by this compound [73].”

Answer: The sentence was improved to facilitate understanding: “On the other hand, at a higher concentration of kaempferol, 100 µM, increments in estrogen concentration were unable to block kaempferol effect, suggesting different pathways for activation by this compound [73].”

Line 216: Plasma (+)-catechin concentrations were also increased…

If it is possible, please, change the reference 96 about tea catechins presented in human plasma after 1-2h from ingestion by another related to wine, that will fit better in this work.

Answer: More details were added about peak plasma of (+)-catechin after ingestion of red wine (Lines 217-220): “Plasma (+)-catechin concentrations increase in response to the ingestion of a single serving of reconstituted red wine. A maximum level of (+)-catechin at 76.7 nmol/L was detected in humans at 1.4 h after intake of both dealcoholized and reconstituted wine” [91].

The paragraph from line 245 to 250 regarding other health effects related to GSE and GSPE, could be shorten and commented in other place in the text. In my opinion, it is not appropiate in the middle of the different results regarding anticancer effects of catechins. This paragraph (but shorten) could fit better in the introduction, or at the beginning or the end of the section 3.2.

Answer: The sentence was summarized and included in the end of section 3.2 (Lines 270-272): “Additionally, other health effects attributed to GSE and GSPE demonstrated potential to improve antioxidant cell defenses and modulate proinflamatory cytokines, witch possible complement the antitumoral functions of this matrixes”.

Lines 320 – 325: Please, improve the link of this paragraph with the chemopreventive effect of delphinidin-3-glucoside by HOTAIR expression inhibition. In the current form is very difficult to understand.

Answer: For better understanding, the sentence was rewritten and a new reference (Hajjari and Salavaty, 2015) was inserted: “HOTAIR, which is over-expressed in different types of cancers, is a lncRNA that plays a role in carcinogenesis and cancer progression by promoting cancer cell viability, growth and metastasis [127]. In its turn, HOTAIR is regulated by the interferon regulatory factor-1 (IRF1) protein, which decreases HOTAIR expression.” (Lines 326-329).

Why the title of Section 5 is only related to wine compounds metabolites and grapes compounds metabolites are not named?

Then, authors speak about grape metabolites in lines 545, 560, 565, 590 etc.

Therefore, I think this section should content information about:

  1. Grape and wine metabolites and breast cancer (in vitro and in vivo studies)

Answer: We appreciate the suggestion and changed the title of this section to “5. Grape and wine metabolites and breast cancer (in vitro and in vivo studies)”.

Line 677: Authors use the abbreviation Cy3G for the first time. However, cyaniding-3-glucoside appears several times before. Please, write the complete name always or, use the abbreviation in parentheses after the full name the first time it appears in the text, and then, continue using the abbreviation the following times it appears.

Answer: Cy3G was modified for cyanidin-3-O-glucoside throughout the text. We also substituted Me-Cy-glucr by methyl-cyanidin-glucuronide and 3`-Me-Cy3glc by 3`-methyl-cyanidin-3-O-glucoside (Line 694).

At line 683, another abbreviation is used.

Answer: Cy3G was modified for cyanidin-3-O-glucoside throughout the text.

Line 683: Remember the use of the past tense in the text.

Answer: The sentence was changed for: “After rats ate cyanidin-3-glucoside, the aglycone was only found in the small intestine, cyanidin-3-glucoside was found in the plasma, and methylated cyanidin-3-glucoside was found in the liver and kidney”.

Line 701: Check references style in the text. “by Fernandes et al. (2013) [221]”

Answer: We consider the commentary and the requested correction were made.

Line 702: some antiproliferative activities…

Answer: We appreciate the suggestion and changed “some antiproliferative activity was changed for some antiproliferative activities” (Line 711-712).

What do you mean with “genuine” anthocyanin?

Answer: Genuine anthocyanin means parental anthocyanin that generates methylated anthocyanins synthetized by the research group of Fernandes et al. 2013. For better comprehension we suggest the modification in the phrase (Lines 711-712): “In a paper by Fernandes et al. [221] the synthetized methylated metabolites still displayed some antiproliferative activities for the three cell lines although not so intense as parental anthocyanin.”

Table 1 collects the metabolites derived from grape and wine compounds after ingestion or after in vitro fermentations. I do not understand why firstly “wine metabolites” are described, followed by a division by different compounds (resveratrol, catechins, anthocyanins and quercetin).

Answer: We agree with this suggestion and the table was modified for a better understand. We firstly described wine or grape metabolites in humans studies, then in animal model and finally in cancer cells. Some data could be included together, as example are the references: 191 and 192, 199 and 200, 204 and 206, 212 and 213, and 226 and 227. Sometimes, wasn`t possible to add the data together because the type of model (human / animal / cancer cell lines) analyzed and the results expressed with different unities, becomes difficult to compare the results.

What is the difference between the catechins described in reference 193 or 197, from 210 or 193? Why are these studies presented separately?

Answer: In reference 193 the authors utilized red wine powder in rats and investigated the metabolites in urine and plasma, they found aromatic acids and hippuric acids besides catechins. In reference 197 the authors utilized red wine in humans and examined the metabolites in plasma. These authors found gallic acid, gallic acid isomers and caffeic acid besides catechins. Therefore, we propose to keep these 2 references in the table. In reference 210 the authors used red wine and dealcoholized red wine in humans. Urines were analyzed and unmethylated and methylated conjugates of catechins were found. So, different metabolites were found in different models.

We think that excluding the different sub-divisions in the table will make it easier for the reader to understand and will not allow duplication of information, as was mentioned as an example by the referee in relation to the reference 193.

Please, reorder the table 1 for a better understanding. Maybe the first section “wine metabolites” is not necessary to avoid repetitions.

Answer: The table was modified for a better understand.

Conclusions should be better written, please, find here some suggestions:

In my opinion, Figure 5 collects the results of this revision. Therefore, this figure must be cited in SECTION 2. Even so, the sentence summarizing the major anticancer activities reported in the review, can be used in the conclusions.

Answer: As suggested, the figure was moved to section 2.

What do you mean with “nanotechnology” in the conclusions? This is not understandable if authors do not describe before novel technologies to improve the bioavailability of phenolic compounds or the bioactive effects of these compounds. Therefore, this note should be better explained if you names “nanotechnology” only as a conclusion.

Answer: We thank the referee for the comment. To avoid misunderstandings, the term “nanotechnology” was removed from the text.

The sentence “It is also important to point out that the effects produced by the glycosidic forms and the aglycones might lead to different routes of absorption and/or metabolization, leading to important variations in bioavailability and global effects produced” could be linked with the following paragraph, where authors described that the dietary SOURCE (wine or grape) and their derived metabolites are responsible for the great variation of effects found, being different factors responsible of these effects, such as the doses used (from XX nM to 100 µM), the qualitative profile of the bioactive compounds presented.

Answer: As suggested, sentences were rewritten: “Based on dietary source, bioactive compounds or their metabolites used in different in vitro and in vivo studies for breast cancer, we conclude that there is a great variation of doses utilized or found. When the studies utilize wine or grape as bioactive compound source, it is possible to observe a great variation on metabolites quality and quantity. On the other hand, when the isolated metabolite or its precursor were used, mainly in cancer cell lines, variations between nM until 100 µM were used, and some authors justify the use of these concentrations to approximate the physiological concentrations. It is also important to point out that the effects produced by the glycosidic forms and the aglycones might lead to different routes of absorption and/or metabolization, leading to important variations in bioavailability and global effects produced.” (Lines 777-785).

It is very important to highlight which are the physiological concentrations used for phenolic compounds and anticancer activities research. If possible, please, add some of this information before conclusions.

Answer: It is important to consider whether the concentrations of bioactive compounds or their metabolites used in different models are nutritional or pharmacological. Usually, the compounds presented in this review, when used as isolated form are utilized in pharmacological doses in clinical trials for breast cancer treatment. Based on the use of functional foods as source of these bioactive compounds, the cancer prevention can be considered and not the chemotherapeutic action, but another important consideration is that dietary components have been shown additive and synergistic effects between then or when associated with chemotherapeutic agents. Our research group published in 2017 (Costa et al., Cancer Chemoprevention by Resveratrol: The p53 Tumor Suppressor Protein as a Promising Molecular Target. Molecules 2017, 22, 1014; doi:10.3390/molecules22061014) one review that discuss the therapeutic perspectives with the use of resveratrol with an emphasis on clinical trials.

We thank the reviewer for the helpful suggestions and comments, which have greatly helped to improve our manuscript.